

# Global LHC constraints on electroweak-inos with SModelS v2.3

Mohammad Mahdi Altakach[1*], Sabine Kraml[1†], Andre Lessa[2‡],
Sahana Narasimha[3,4°], Timothée Pascal[1§], Théo Reymermier[5¶]
and Wolfgang Waltenberger[3,4||]

**1** Laboratoire de Physique Subatomique et de Cosmologie (LPSC), Université Grenoble-Alpes,
CNRS/IN2P3, 53 Avenue des Martyrs, F-38026 Grenoble, France
**2** Centro de Ciências Naturais e Humanas, Universidade Federal do ABC,
Santo André, 09210-580 SP, Brazil
**3** Institut für Hochenergiephysik, Österreichische Akademie der Wissenschaften,
Nikolsdorfer Gasse 18, A-1050 Wien, Austria
**4** University of Vienna, Faculty of Physics, Boltzmanngasse 5, A-1090 Wien, Austria
**5** Université de Lyon, Université Claude Bernard Lyon 1, CNRS/IN2P3,
Institut de Physique des 2 Infinis de Lyon, UMR 5822, F-69622, Villeurbanne, France

* altakach@lpsc.in2p3.fr , † sabine.kraml@lpsc.in2p3.fr , ‡ andre.lessa@ufabc.edu.br ,
° sahana.narasimha@oeaw.ac.at , § timothee.pascal@lpsc.in2p3.fr ,
¶ t.reymermier@ip2i.in2p3.fr , || walten@hephy.oeaw.ac.at

## Abstract

Electroweak-inos, superpartners of the electroweak gauge and Higgs bosons, play a special role in supersymmetric theories. Their intricate mixing into chargino and neutralino mass eigenstates leads to a rich phenomenology, which makes it difficult to derive generic limits from LHC data. In this paper, we present a global analysis of LHC constraints for promptly decaying electroweak-inos in the context of the minimal supersymmetric standard model, exploiting the SMODELS software package. Combining up to 16 ATLAS and CMS searches for which electroweak-ino efficiency maps are available in SMODELS, we study which combinations maximise the sensitivity in different regions of the parameter space, how fluctuations in the data in individual analyses influence the global likelihood, and what is the resulting exclusion power of the combination compared to the analysis-by-analysis approach.

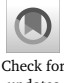

# 1  Introduction

The quest for new physics is at a crossroads. While experimental searches at the Large Hadron Collider (LHC) are still being pursued final state by final state, targeting specific (yet often unrealistic) simplified-model scenarios, global reinterpretations of the wealth of experimental results become more and more important in order to understand which scenarios are really excluded and where new physics may still be hiding. This is particularly true for searches for supersymmetry (SUSY) [1,2], as SUSY phenomenology is extremely rich and realistic scenarios are expected to give signals in several different channels. Even in the simplest incarnation, the Minimal Supersymmetric Standard Model (MSSM), decay patterns, and thus signatures, can be quite complex. A global approach is then required to elucidate regions of the parameter space yet uncovered. The question is a burning one at this stage of LHC Run 3, as new analysis strategies may be in order to cover existing holes.

A sector of the MSSM that is of peculiar interest in this regard is that of the electroweak-inos (EW-inos) — mixtures of the superpartners of the electroweak gauge and Higgs bosons of the Standard Model (SM). Electroweak-inos play a crucial role in gauge coupling unification in the MSSM [3–5], they are intimately tied to an understanding of the electroweak scale through the Higgs mixing parameter $\mu$ [6,7] (see also [8]), and they are of primordial interest in the context of SUSY dark matter [9]. Even if all the rest of the SUSY spectrum is decoupled, light EW-inos are a well-motivated and hugely interesting possibility for beyond the SM (BSM) physics.

In this paper, we perform a global analysis of LHC constraints for promptly decaying EW-inos. To this end, we exploit the SMODELS [10–14] software package and its large database of experimental results from Run 1 and Run 2 of the LHC. SMODELS is a public tool for the fast reinterpretation of LHC searches for new physics on the basis of simplified-model results. Its working principle is to decompose the signatures of full theoretical scenarios into simplified-model components, which are then confronted against the experimental constraints from a large database of results. The approach assumes that the kinematic distributions for the tested BSM scenario are approximately the same as for the simplified models assumed in the experimental results, so that the same signal acceptances apply.

The combination of (approximately) independent analyses in SMODELS through a global likelihood was introduced in a recent paper [14], where we also considered the EW-ino sector as a show-case example. While in [14] only two analyses, one from ATLAS and one from CMS, were combined, we here go a significant step further and dynamically determine the best (i.e., the most sensitive) combination from a set of 16 analyses (11 from ATLAS and 5 from CMS), for which likelihoods for EW-inos can be computed. The procedure relies on the combiner algorithm developed in [15] for the search for possible dispersed signals in the LHC results.

A global likelihood from the combination of different individual analyses is relevant for two reasons. First, the signal of a particular BSM scenario may be manifest in different final states, which are constrained by different analyses. Combining them uses more of the available data and thus provides more robust (and usually stronger) constraints. In any case, the sensitivity of a combination of relevant analyses is always bigger than the sensitivity of any single analysis taken alone. Second, experimental analyses can always be subject to over- or under-fluctuations in the data. In the former case, the observed limit is weaker, in the latter case stronger, than the expected limit. Again, the combination of different, approximately independent analyses can mitigate this effect and provide more robust constraints.

Global EW-ino analyses were also performed by the GAMBIT collaboration, in [16] for the case of a neutralino as the lightest SUSY particle (LSP) and in [17] for the case of a gravitino LSP. The GAMBIT approach is different from ours in several ways. In particular, GAMBIT aims at maximizing the profile likelihood in a global fit, taking into account also other constraints than those from LHC searches. Regarding LHC searches, GAMBIT performs a simulation-based reinterpretation (see [18] for a review of the different methods). Regarding the set of experimental analysis, [16] studied the impact of ATLAS and CMS EW-ino searches for 36 fb$^{-1}$ of 13 TeV LHC data. In contrast, [17] considered 15 ATLAS and 12 CMS searches at 13 TeV, along with a number of ATLAS and CMS measurements of SM signatures.

In this paper, we pursue a different objective. Instead of asking which region of EW-ino parameter space best fits the data, we investigate in depth how the current EW-ino search results constrain this sector of the MSSM. To this end, we study which combinations maximise the sensitivity in different regions of parameter space, how fluctuations in the data observed in individual analyses influence the global likelihood, and what is the resulting exclusion power of the combination compared to the analysis-by-analysis approach. The set of experimental analyses used consists of 16 ATLAS and CMS publications (3 from Run 1 and 13 from Run 2), 9 of which are for full Run 2 luminosity.

The paper is structured as follows. In Section 2, we first review the EW-ino sector of the MSSM, discussing masses and mixings as well as decay patterns and signatures. In Section 3, we present the relevant analyses in the SMODELS database, which we consider in our study. In Section 4, we discuss details of the numerical analysis, including the scan over the EW-ino parameter space and the strategy for determining the set of analyses to combine for each parameter point. This brings us to Section 5, where we present and discuss our results. Section 6 contains a brief summary and conclusions. Two appendices complete the paper. Appendix A presents the new SMODELS interface to RESUMMINO [19–21] for computing electroweak SUSY cross sections beyond leading order, and Appendix B discusses the relevance of the mass compression parameter (`minmassgap`) in SMODELS for our results.

Throughout the paper, it is assumed that the reader is reasonably familiar with the concepts of SMODELS. If this is not the case, we refer to the original SMODELS paper [10] and the extensive online manual for a detailed introduction. The calculation of likelihoods in SMODELS is reviewed in [14].

## 2 The electroweak-ino sector of the MSSM

In this section, we briefly review the EW-ino sector of the R-parity and CP-conserving MSSM, clarifying our notation and conventions. Moreover, we discuss decay patterns and resulting LHC signatures for a couple of specific scenarios. Throughout the paper we assume that the other SUSY particles are heavier than the EW-inos, so they do not influence the phenomenology discussed here.

### 2.1 Neutralino and chargino masses

We denote neutralinos and charginos as $\tilde{\chi}^0_{1\ldots4}$ and $\tilde{\chi}^\pm_{1,2}$, respectively. They are the mass eigenstates that form from the higgsinos and electroweak gauginos (the fermionic superpartners of the Higgs and electroweak gauge bosons, respectively) due to the effects of electroweak symmetry breaking. Concretely, neutralinos are linear combinations of the bino $\widetilde{B}$, neutral wino $\widetilde{W}^0$ and neutral higgsino $\widetilde{H}^0_u$ and $\widetilde{H}^0_d$ gauge eigenstates, while charginos are linear combinations of the charged wino ($\widetilde{W}^+$ and $\widetilde{W}^-$) and charged higgsino ($\widetilde{H}^+_u$ and $\widetilde{H}^-_d$) gauge eigenstates.

In the basis $\psi^0 = (\widetilde{B}, \widetilde{W}^0, \widetilde{H}^0_d, \widetilde{H}^0_u)^T$, $\psi^\pm = (\widetilde{W}^+, \widetilde{H}^+_u, \widetilde{W}^-, \widetilde{H}^-_d)^T$, the relevant part of the Lagrangian is [8, 22]

$$\mathcal{L}_{\text{mass}} = -\frac{1}{2}(\psi^0)^T \mathcal{M}_N \psi^0 - \frac{1}{2}(\psi^\pm)^T \mathcal{M}_C \psi^\pm + \text{c.c.}, \tag{1}$$

where

$$\mathcal{M}_N = \begin{pmatrix} M_1 & 0 & -c_\beta\, s_W\, m_Z & s_\beta\, s_W\, m_Z \\ 0 & M_2 & c_\beta\, c_W\, m_Z & -s_\beta\, c_W\, m_Z \\ -c_\beta\, s_W\, m_Z & c_\beta\, c_W\, m_Z & 0 & -\mu \\ s_\beta\, s_W\, m_Z & -s_\beta\, c_W\, m_Z & -\mu & 0 \end{pmatrix}, \tag{2}$$

and

$$\mathcal{M}_C = \begin{pmatrix} \mathbf{0} & \mathbf{X}^T \\ \mathbf{X} & \mathbf{0} \end{pmatrix}, \qquad \mathbf{X} = \begin{pmatrix} M_2 & \sqrt{2}\, s_\beta\, m_W \\ \sqrt{2}\, c_\beta\, m_W & \mu \end{pmatrix}, \tag{3}$$

are the neutralino and chargino mass matrices. Here, $M_1$, $M_2$ and $\mu$ are the bino, wino and higgsino mass parameters, respectively, and we have introduced the abbreviations $s_\beta = \sin\beta$, $c_\beta = \cos\beta$, $s_W = \sin\theta_W$, and $c_W = \cos\theta_W$, with $\theta_W$ the weak mixing angle and $\tan\beta = v_u/v_d$ the ratio of the Higgs vacuum expectation values. The chargino mass matrix is written in $2\times2$ block form for convenience.

The mass eigenstates are related to the gauge eigenstates by the unitary matrices $N$, $U$ and $V$, which diagonalise the mass matrices:

$$N^* \mathcal{M}_N N^{-1} = \text{diag}\left(m_{\tilde{\chi}^0_1}, \ldots, m_{\tilde{\chi}^0_4}\right), \tag{4}$$

$$U^* \mathbf{X} V^{-1} = \text{diag}\left(m_{\tilde{\chi}^\pm_1}, m_{\tilde{\chi}^\pm_2}\right), \tag{5}$$

so that

$$\tilde{\chi}^0_i = N_{ij}\psi^0_j, \qquad \begin{pmatrix} \tilde{\chi}^+_1 \\ \tilde{\chi}^+_2 \end{pmatrix} = V\begin{pmatrix} \widetilde{W}^+ \\ \widetilde{H}^+_u \end{pmatrix}, \qquad \begin{pmatrix} \tilde{\chi}^-_1 \\ \tilde{\chi}^-_2 \end{pmatrix} = U\begin{pmatrix} \widetilde{W}^- \\ \widetilde{H}^-_d \end{pmatrix}. \tag{6}$$

By convention, the physical states are mass ordered: $|m_{\tilde{\chi}^0_i}| < |m_{\tilde{\chi}^0_j}|$ for $i < j$ and $m_{\tilde{\chi}^\pm_1} < m_{\tilde{\chi}^\pm_2}$. Following the SUSY Les Houches Accord (SLHA) [23] conventions, in the absence of CP violation we choose the mixing matrices to be real, allowing for *signed* (negative) mass eigenvalues for the neutralinos. The lightest neutralino, $\tilde{\chi}^0_1$, is typically also the Lightest Supersymmetric Particle (LSP) and the dark matter candidate.

If the mass parameters $M_1$, $M_2$ and $\mu$ in eqs. (2) and (3) are sufficiently different from each other, the mixing is small and one ends up with a bino-like neutralino with mass of about $M_1$, an almost mass-degenerate pair of wino-like chargino/neutralino with mass of about $M_2$, and a triplet of higgsino-like neutralinos and chargino with mass of about $\mu$.[1] The bino, wino and higgsino contents of neutralino $\tilde{\chi}_i^0$ are given by $|N_{i1}|^2$, $|N_{i2}|^2$ and $|N_{i3}|^2 + |N_{i4}|^2$, respectively. Likewise, the wino and higgsino admixtures of chargino $\tilde{\chi}_i^+$ are given by $|V_{i1}|^2$ and $|V_{i2}|^2$.

In practice, loop corrections are important, and the masses and mixing angles of the neutralinos and charginos are best computed numerically. In this study, we use SOFTSUSY [24,25] for this purpose, which also provides the decay tables.

## 2.2 LHC signatures

Let us next turn to the collider signatures expected from EW-ino production at the LHC. To this end, we need to consider the production cross sections and decay patterns for different scenarios. The production cross sections at the 13 TeV LHC are shown in Figure 1, assuming pure wino or pure higgsino states with degenerate masses (the denotation as $\tilde{\chi}_1^\pm$, $\tilde{\chi}_1^0$, etc. is completely arbitrary). As can be seen, winos are produced much more copiously than higgsinos of the same mass. The production of pure binos, on the other hand, is not shown on the plot, as it is negligible compared to the other modes. For the discussion of signatures, we therefore focus on decays of wino-like or higgsino-like EW-inos into lighter ones. Figure 2 shows examples of four scenarios with different LHC signatures which are discussed below.

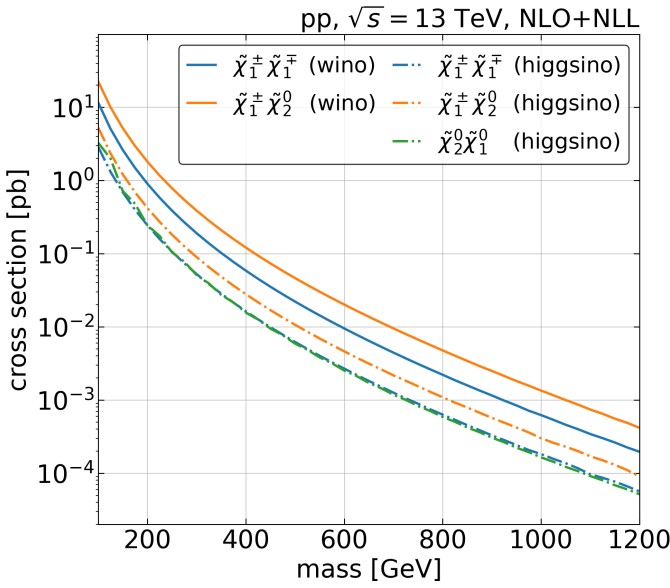

Figure 1: Production cross sections of pure wino or higgsino states with degenerate masses (values taken from [26]). The production cross sections for higgsino-like $\tilde{\chi}_1^\pm \tilde{\chi}_2^0$ and $\tilde{\chi}_2^0 \tilde{\chi}_1^0$ are almost identical.

---

[1]While in principle there can be a relative phase between the -ino mass parameters, for simplicity we consider only non-negative values ($M_1, M_2, \mu \geq 0$) in our study.

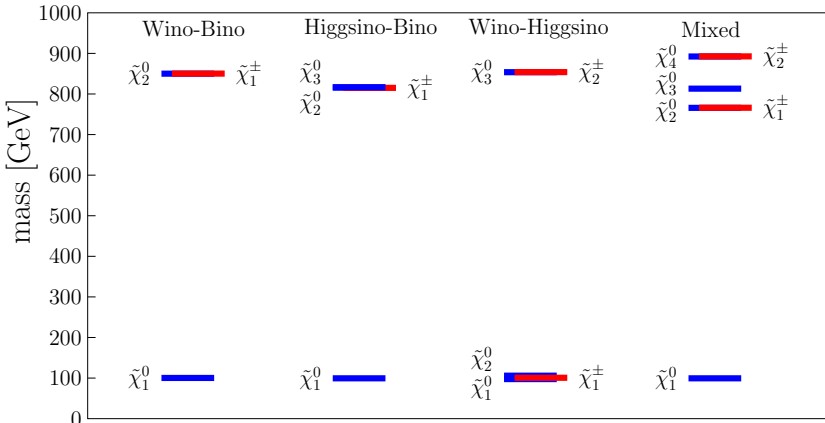

Figure 2: Illustrative spectra for the different physics scenarios discussed in Section 2.2. The mass parameters that set the (heavy, light) scales are, for wino-bino: $(M_2, M_1)$, for higgsino-bino: $(\mu, M_1)$, for wino-higgsino: $(M_2, \mu)$; for the mixed scenario, $(M_2 = \mu, M_1)$ was used. The 'heavy' scale is always set to 800 GeV, and the 'light' scale to 100 GeV. In the first three scenarios, the third mass parameter is decoupled. To guide the eye, neutralinos are indicated in blue and charginos in red.

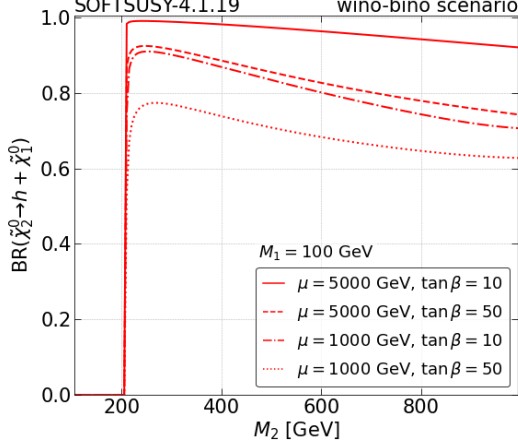

Figure 3: Decay branching ratios of a wino-like $\tilde{\chi}_2^0$ into a bino-like $\tilde{\chi}_1^0$ plus SM-like Higgs boson $h$ as a function of $M_2$, for $M_1 = 100$ GeV and different values of $\tan\beta$ and $\mu$. Here, $\mathrm{BR}(\tilde{\chi}_2^0 \to Z + \tilde{\chi}_1^0) = 1 - \mathrm{BR}(\tilde{\chi}_2^0 \to h + \tilde{\chi}_1^0)$.

**Wino-bino scenario**

The case most commonly considered in LHC studies is the production of wino-like $\tilde{\chi}_1^\pm$ and $\tilde{\chi}_2^0$ (with $m_{\tilde{\chi}_1^\pm} = m_{\tilde{\chi}_2^0}$) which decay to a bino-like $\tilde{\chi}_1^0$. This is realised for $\mu \gg M_2 > M_1$; we call this the wino-bino scenario. An example is shown in the left-most spectrum in Figure 2. In the absence of intermediate sfermions, the available 2-body decay modes of the $\tilde{\chi}_1^\pm$ and $\tilde{\chi}_2^0$ are

$$\tilde{\chi}_1^\pm \to W^\pm + \tilde{\chi}_1^0, \tag{7}$$

$$\tilde{\chi}_2^0 \to Z + \tilde{\chi}_1^0, \quad \text{or} \quad h + \tilde{\chi}_1^0, \tag{8}$$

where $h$ is the SM-like Higgs boson with a mass of 125 GeV (the other Higgs bosons are assumed to be heavy). If the 2-body decays are kinematically forbidden, $\tilde{\chi}_1^\pm \to f\bar{f}' + \tilde{\chi}_1^0$ and $\tilde{\chi}_2^0 \to f\bar{f} + \tilde{\chi}_1^0$ via off-shell $W$ and $Z$ respectively. The signals looked for are thus primarily $W^\pm Z + \not{E}_T$ and $W^\pm h + \not{E}_T$ from $pp \to \tilde{\chi}_1^\pm \tilde{\chi}_2^0$ production, as well as $W^+ W^- + \not{E}_T$ from $pp \to \tilde{\chi}_1^+ \tilde{\chi}_1^-$

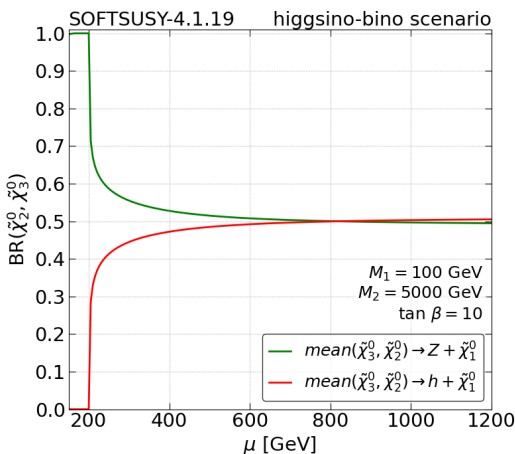

Figure 4: Decay branching ratios of higgsino-like $\tilde{\chi}^0_{2,3}$ into a bino-like $\tilde{\chi}^0_1$ plus a Higgs or $Z$ boson as a function of $\mu$, for $M_1 = 100$ GeV, $M_2 = 5000$ GeV and $\tan\beta = 10$. Since the $\tilde{\chi}^0_{2,3}$ are indistinguishable, only the total rates of decays into $h + \tilde{\chi}^0_1$ and $Z + \tilde{\chi}^0_1$ are shown.

production. Other modes, like $pp \to \tilde{\chi}^0_2\tilde{\chi}^0_2$, $\tilde{\chi}^\pm_1\tilde{\chi}^0_1$, $\tilde{\chi}^0_2\tilde{\chi}^0_1$ production, are less important, partly because of smaller production cross sections and partly because the resulting signals suffer from larger SM background. The $W$ and $Z$ bosons are typically looked for in leptonic final states ($W \to \ell\nu$, $Z \to \ell^+\ell^-$), while Higgs bosons are identified through $b\bar{b}$ or $\gamma\gamma$ final states. Only very recently, for the full Run 2 dataset, analyses were carried out in fully hadronic final states using boosted boson tagging [27, 28].

The $\tilde{\chi}^\pm_1$ decay into $W^\pm\tilde{\chi}^0_1$ (or $f\bar{f}'\tilde{\chi}^0_1$) proceeds through the left current and, as the only available decay mode, has 100% branching ratio. The branching ratios of the $\tilde{\chi}^0_2$ decays, however, depend on the details of the scenario. While BR($\tilde{\chi}^0_2 \to Z\tilde{\chi}^0_1$) $\simeq 1$ for $m_{\tilde{\chi}^0_2} - m_{\tilde{\chi}^0_1} < m_h$, the decay into Higgs bosons quickly becomes dominant once kinematically allowed. The exact branching ratios depend also on the values of $\mu$ and $\tan\beta$, since neutralinos couple to $Z$ bosons only through their higgsino components.[2] For illustration, Figure 3 shows the decay branching ratios of a wino-like $\tilde{\chi}^0_2$ into a SM-like Higgs boson and a bino-like $\tilde{\chi}^0_1$ as a function of $M_2 \simeq m_{\tilde{\chi}^0_2}$, for $m_{\tilde{\chi}^0_1} \simeq M_1 = 100$ GeV and varying values of $\tan\beta$ and $\mu$. We see from this figure that limits under the assumption of 100% signal in either $\tilde{\chi}^\pm_1\tilde{\chi}^0_2 \to W^\pm Z + \not{E}_T$ or $\tilde{\chi}^\pm_1\tilde{\chi}^0_2 \to W^\pm h + \not{E}_T$ are unrealistic for most of the parameter space. Even for the $W^\pm h + \not{E}_T$ channel, for which the difference in signal compared to the simplified model result can be small, the combination of results with the $W^\pm Z + \not{E}_T$ channel should be interesting. Moreover, $\tilde{\chi}^+_1\tilde{\chi}^-_1 \to W^+W^- + \not{E}_T$ is always present in addition and needs to be included for realistic constraints.

**Higgsino-bino scenario**

For $M_2 \gg \mu > M_1$, the next-to-lightest mass scale is set by the higgsinos, with the lightest state being bino-like, as illustrated by the 2nd spectrum from the left in Figure 2. In this higgsino-bino scenario, the EW-ino signatures arise from $pp \to \tilde{\chi}^+_1\tilde{\chi}^-_1$, $\tilde{\chi}^\pm_1\tilde{\chi}^0_{2,3}$ and $\tilde{\chi}^0_2\tilde{\chi}^0_3$ production followed by

$$\tilde{\chi}^\pm_1 \to W^\pm + \tilde{\chi}^0_1, \tag{9}$$

$$\tilde{\chi}^0_{2,3} \to Z + \tilde{\chi}^0_1, \quad \text{or} \quad h + \tilde{\chi}^0_1. \tag{10}$$

---

[2]Concretely, the $\tilde{\chi}^0_i\tilde{\chi}^0_j Z$ coupling is given by $\frac{ig}{2c_W}(N_{i4}N^\star_{j4} - N_{i3}N^\star_{j3})$, see [22, 29].

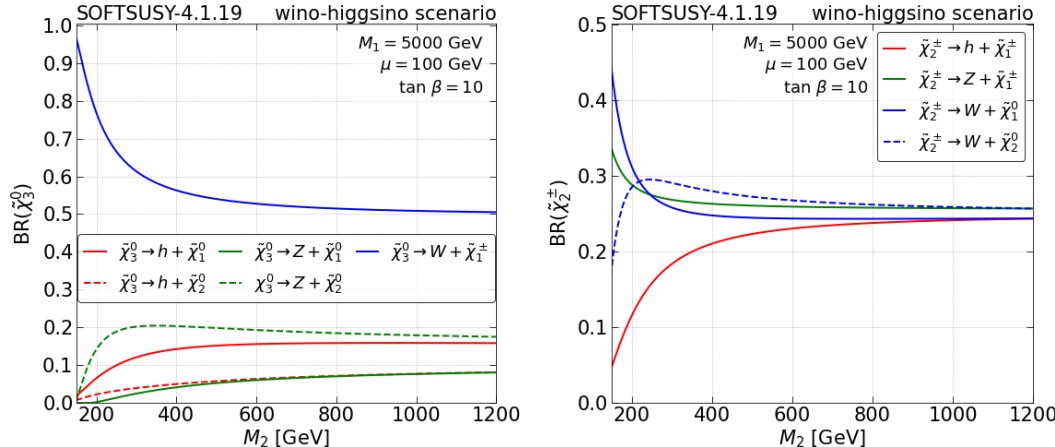

Figure 5: Decay branching ratios of $\tilde{\chi}_3^0$ (left) and $\tilde{\chi}_2^\pm$ (right) in the wino-higgsino scenario as a function of $M_2$, for $\mu = 100$ GeV, $M_1 = 5000$ GeV, and $\tan\beta = 10$.

The $\tilde{\chi}_{2,3}^0$ are practically degenerate in mass. For large enough $\mu$, one of them decays dominantly (roughly to 70–80%) into $Z\tilde{\chi}_1^0$, while the other decays dominantly into $h\tilde{\chi}_1^0$ (also roughly to 70–80%), giving almost equal rates of $Z$ and $h$. The total EW-ino signal is therefore again a mix of $WW$, $WZ$, $Wh$ and $Zh + \not{E}_T$ signatures, with a small addition of $ZZ$ and $hh + \not{E}_T$. The relative rates of $Z$ or $h$ bosons from $\tilde{\chi}_{2,3}^0$ decays are shown in Figure 4; they are almost insensitive to variations of $M_2$ and $\tan\beta$.

**Wino-higgsino scenario**

If the LSP is higgsino-like instead of bino-like, things become much more complicated. The wino-higgsino scenario is realised for $M_1 \gg M_2 > \mu$. In this case, the $\tilde{\chi}_{1,2}^0$ and $\tilde{\chi}_1^\pm$ are a triplet of higgsino-like states with $m_{\tilde{\chi}_1^0} < m_{\tilde{\chi}_1^\pm} < m_{\tilde{\chi}_2^0}$. The mass differences between them are generally small, $m_{\tilde{\chi}_1^\pm} - m_{\tilde{\chi}_1^0} \lesssim$ few GeV, and $m_{\tilde{\chi}_2^0} - m_{\tilde{\chi}_1^0}$ is in the range of about 3–30 GeV, see the 3rd spectrum from the left in Figure 2. The wino-like states are now the $\tilde{\chi}_2^\pm$ and $\tilde{\chi}_3^0$, and they have a large variety of possible decay modes:

$$\tilde{\chi}_2^\pm \to W^\pm \tilde{\chi}_2^0, \quad Z\tilde{\chi}_1^\pm, \quad h\tilde{\chi}_1^\pm, \quad W^\pm \tilde{\chi}_1^0, \tag{11}$$

$$\tilde{\chi}_3^0 \to Z\tilde{\chi}_2^0, \quad h\tilde{\chi}_2^0, \quad W^\mp\tilde{\chi}_1^\pm, \quad Z\tilde{\chi}_1^0, \quad h\tilde{\chi}_1^0, \tag{12}$$

followed by $\tilde{\chi}_2^0 \to f\bar{f}'\tilde{\chi}_1^\pm, f\bar{f}\tilde{\chi}_1^0$ and/or $\tilde{\chi}_1^\pm \to f\bar{f}'\tilde{\chi}_1^0$ transitions if the $\tilde{\chi}_2^\pm$ or $\tilde{\chi}_3^0$ decay is not directly into the LSP. Since $pp \to \tilde{\chi}_3^0\tilde{\chi}_3^0$ production is suppressed, only $pp \to \tilde{\chi}_2^\pm\tilde{\chi}_3^0$ and $pp \to \tilde{\chi}_2^+\tilde{\chi}_2^-$ are relevant and lead to a complicated mix of $W^+W^-$, $W^\pm Z$, $W^\pm h$, $ZZ$, $Zh$, $hh$ plus $\not{E}_T$ final states, often accompanied by additional soft jets or leptons.

The relevant branching ratios are depicted in Figure 5 as a function of the wino mass parameter, for $\mu = 100$ GeV, $M_1 = 5000$ GeV and $\tan\beta = 10$. Variations with $M_1$ and $\tan\beta$ are small and do not change the overall picture of a multitude of relevant final states. Indeed the EW-ino signal is split almost democratically into the different di-boson+$\not{E}_T$ and di-boson+$X$+$\not{E}_T$ channels, $X$ meaning additional soft jets and/or leptons. We can therefore expect that any individual simplified model result will be too weak to significantly constrain this scenario; the statistical combination of results from different analyses, each constraining the EW-ino signal in a different final state, should be of great advantage for obtaining sensitive limits.[3]

---

[3]The exception are perhaps constraints from the fully hadronic searches [27,28], which are sensitive to a variety of boosted boson final states.

**Wino LSP**

When $M_2$ is the smallest mass parameter, we have a doublet of wino-like $\tilde{\chi}_1^0$ and $\tilde{\chi}_1^\pm$ as the lightest states. The $\tilde{\chi}_1^0$ is still the LSP, but the mass difference to the $\tilde{\chi}_1^\pm$ is so small (only about 150–160 MeV) that the $\tilde{\chi}_1^\pm$ becomes long-lived on collider scales. The signatures to look for are then 1–2 disappearing tracks from $\tilde{\chi}_1^\pm \tilde{\chi}_1^0$ and $\tilde{\chi}_1^+ \tilde{\chi}_1^-$ production, respectively. While disappearing track analyses are included in the SMODELS database, they are very different from the analyses searching for promptly decaying EW-inos and, if relevant, their constraints are completely dominant. We therefore do not consider (pure) wino LSPs in our analysis.

**Mixed scenarios**

If two (or all three) EW-ino mass parameters are close to each other, we obtain mixed scenarios, in which the properties of the relevant charginos and neutralinos are not well approximated as being bino, wino or higgsino. One consequence of such mixing is that more states are relevant at the same time and they share out their couplings. Moreover, the mass splittings among similar states increase.

For example, for $M_1 = 100$ GeV and $M_2 = \mu = 800$ GeV (with $\tan\beta = 10$), we have a bino-like $\tilde{\chi}_1^0$ of mass $m_{\tilde{\chi}_1^0} = 105$ GeV; the heavier states have masses of $m_{\tilde{\chi}_{2,3,4}^0} = 787, 824,$ 922 GeV and $m_{\tilde{\chi}_{1,2}^\pm} = 787, 922$ GeV with the $\tilde{\chi}_2^0$ ($\tilde{\chi}_4^0$) being 42% (58%) wino and 58% (42%) higgsino, see the right-most spectrum in Figure 2. The charginos are also highly mixed, having both wino and higgsino components. Only the $\tilde{\chi}_3^0$ is almost pure higgsino. Consequently, the production of all $\tilde{\chi}_{1,2}^\pm$ and $\tilde{\chi}_{2,3,4}^0$ combinations is important and adds to the complexity of the EW-ino signal. In particular, while the $\tilde{\chi}_1^\pm$ and $\tilde{\chi}_{2,3}^0$ decay directly into the $\tilde{\chi}_1^0$ (plus $W$, $h$ or $Z$ bosons), the $\tilde{\chi}_2^\pm$ decays to 44% into $W^\pm \tilde{\chi}_2^0$, 31% into $Z \tilde{\chi}_1^\pm$ and 5% into $h \tilde{\chi}_1^\pm$, only 12% going into $W^\pm \tilde{\chi}_1^0$. Likewise, the $\tilde{\chi}_4^0$ decays to almost 80% into $W^\pm \tilde{\chi}_1^\mp$; only about 10% of the $\tilde{\chi}_4^0$ decays go into $h \tilde{\chi}_1^0$ and 4% into $Z \tilde{\chi}_1^0$.

In a similar fashion, when the mass scale of the LSP is set by two parameters with very similar values, e.g. $M_2 \simeq M_1 \ll \mu$ or $\mu \simeq M_1 \ll M_2$, this leads to quite complicated scenarios. The signatures can be similar to those of the wino-higgsino case discussed above, but with more variety of nearly-degenerate states. We do not go into more details here, but only note that $M_1 \simeq M_2$ can give a mostly wino-like LSP with a large enough mass splitting such that the $\tilde{\chi}_1^\pm$ decays promptly.

## 3 Relevant analyses in the database

In this section, we give an overview of the experimental results in the SMODELS v2.3 [14] database that enter our study. In particular, we explain which material is provided by the experimental collaborations and how it is used in SMODELS. A summary is provided in Table 1. Since our aim is to build global likelihoods from these data, we consider efficiency map (EM) type results [11] only. Particular attention is given to the combination of signal regions (SRs) through full [30] or simplified [31, 32] statistical models. In the following, 'lepton' refers to electrons or muons ($\ell = e, \mu$) unless stated otherwise.

### 3.1 ATLAS results

**ATLAS-SUSY-2013-11** [33]: The 20.3 fb$^{-1}$ Run 1 search for charginos, neutralinos and sleptons decaying into final states with two opposite-sign leptons, 0 or $\geq 2$ jets, and $\not{E}_T$. The analysis is divided into 13 SRs, six targeting the production of sleptons, six targeting pair-production of charginos and one targeting chargino-neutralino production. Only the latter

Table 1: List of EW-ino analyses from LHC Run 1 ($\sqrt{s} = 8$ TeV) and Run 2 (13 TeV) considered in this study, "lumi" giving the integrated luminosity in fb$^{-1}$. The column "comb." specifies whether and how SRs are combined: "pyhf" means a full or SIMPLIFY'ed HISTFACTORY model is used through interface with PYHF; SLv1 (SLv2) means that a covariance matrix is used in the Simplified Likelihood scheme of [32] ([47]); a – means that only the best SR is used.

| ID | Run | lumi | Final State (+$\not{E}_T$) | EMs (+$\not{E}_T$) | SRs | comb. |
|---|---|---|---|---|---|---|
| ATLAS-SUSY-2013-11 [33] | 1 | 20.3 | 2 lept., 0 or $\geq$ 2 jets, 0$b$ | $WW^{(*)}$ | 13 | – |
| ATLAS-SUSY-2013-12 [34] | 1 | 20.3 | 3 lept. (0–2 $\tau$'s), 0$b$ | $WZ^{(*)}, Wh$ | 2 | – |
| ATLAS-SUSY-2016-24 [35] | 2 | 36.1 | 2–3 lept., 0 or $\geq$ 2 jets, 0$b$ | $WZ$ | 9 | – |
| ATLAS-SUSY-2017-03 [36] | 2 | 36.1 | 2–3 lept., 0 or $\geq$ 1 jets, 0$b$ | $WZ$ | 8 | – |
| ATLAS-SUSY-2018-05 [37] | 2 | 139 | 2 lept., $\geq$ 1 jets | $WZ^{(*)}$ | 13 | pyhf |
| ATLAS-SUSY-2018-06 [38] | 2 | 139 | 3 lept., 0 or 1–3 jets, 0$b$ | $WZ^{(*)}$ | 2 | – |
| ATLAS-SUSY-2018-32 [39] | 2 | 139 | 2 lept., 0 or 1 jets, 0$b$ | $WW$ | 36 | pyhf |
| ATLAS-SUSY-2018-41 [27] | 2 | 139 | 4 jets or 2$b$ + 2 jets, 0 lept. | $WW, WZ, Wh,$ $Zh, ZZ, hh$ | 3 | SLv1 |
| ATLAS-SUSY-2019-02 [40] | 2 | 139 | 2 lept., 0 or 1 jets, 0$b$ | $WW$ | 24 | SLv1 |
| ATLAS-SUSY-2019-08 [41] | 2 | 139 | 1 lept., ($h \to$)$b\bar{b}$ | $Wh$ | 9 | pyhf |
| ATLAS-SUSY-2019-09 [42] | 2 | 139 | 3 lept., 0 or $\geq$ 1 jets, 0$b$ | $WZ^{(*)}$ | 20+31 | pyhf |
| CMS-SUS-13-012 [43] | 1 | 19.5 | 0 lept., $\geq$ 3 jets ($q$ or $b$) | $WW, WZ, ZZ$ | 36 | – |
| CMS-SUS-16-039 [44] | 2 | 35.9 | 2+ lept., 0–2 hadr. $\tau$'s, 0$b$ | $WZ^{(*)}$ | 11 | SLv1 |
| CMS-SUS-16-048 [45] | 2 | 35.9 | 2 soft lept., $\geq$ 1 jets, 0$b$ | $WZ^{*}$ | 12+9 | SLv1 |
| CMS-SUS-20-004 [46] | 2 | 137 | 0 lept., 2$h(\to b\bar{b})$ | $hh$ | 22 | SLv2 |
| CMS-SUS-21-002 [28] | 2 | 137 | $\geq$ 2 AK8 jets, 0 or $\geq$ 1$b$'s 0 lept. | $WW, WZ, Wh$ | 35 | SLv1 |

considers jets. Implemented in the SMODELS database are EMs for slepton and chargino production, produced through the MADANALYSIS 5 recast [48]. Since no correlation information was provided by the experimental collaboration, SMODELS only uses the most sensitive (a.k.a. "best") SR.

**ATLAS-SUSY-2013-12 [34]:** The 20.3 fb$^{-1}$ Run 1 search for chargino-neutralino pair production decaying into three leptons ($e$, $\mu$ or $\tau$) and $\not{E}_T$. At least one electron or muon is required among the three leptons. The ATLAS collaboration published on HEPDATA the EMs of 4 SRs [49], each of them targeting a specific ($WZ$, $Wh$, $\tilde{e}/\tilde{\mu}$ or $\tilde{\tau}$-mediated) scenario. These 4 EMs are implemented in the SMODELS database; no correlation information being available, only the best SR is used.

The SRs of the two analyses above do not overlap, since one requires exactly 2 leptons and the other one exactly 3.

**ATLAS-SUSY-2016-24 [35]:** A search in final states with exactly two or three leptons plus $\not{E}_T$, targeting the electroweak production of charginos, neutralinos and sleptons. The analysis is based on 36.1 fb$^{-1}$ of data from Run 2. Its three search regions (2$\ell$ + 0 jets, 2$\ell$ + jets and 3$\ell$), are split into a total of 37 SRs, 28 exclusive and 9 inclusive ones. Acceptance and efficiency maps are available on HEPDATA [50] for the 9 inclusive 2$\ell$ SRs and the 11 exclusive 3$\ell$ SRs and implemented in SMODELS. Relevant for EW-inos decaying via SM gauge bosons (as opposed

to decays via light sleptons) are 9 SRs: the 3 inclusive $2\ell$+jets SRs and 6 of the exclusive $3\ell$ SRs. No correlation information is provided for the exclusive SRs. However, the most sensitive (the "best") SR reproduces quite well the official limit for $pp \to \tilde{\chi}_1^\pm \tilde{\chi}_2^0 \to WZ + \not{E}_T$ from this analysis. The observed limit in this analysis is about $1\sigma$ stronger than the expected one.

**ATLAS-SUSY-2017-03 [36]:** Another search in two-lepton and three-lepton final states based on 36.1 fb$^{-1}$ of Run 2 data. The main difference to ATLAS-SUSY-2016-24 above is that it uses a recursive jigsaw reconstruction. Moreover, it targets only chargino-neutralino pair production with decays via $W/Z$ bosons. It considers two types of search regions, one with two leptons and at least two jets ($2\ell$ category), and one with three leptons and up to three jets ($3\ell$ category). In total, the analysis has 8 SRs, for all of which EMs for the $pp \to \tilde{\chi}_1^\pm \tilde{\chi}_2^0 \to WZ + \not{E}_T$ simplified model are available on HEPDATA [51] and implemented in SMODELS. Lacking an explicit statistical model, SMODELS uses only the best SR for the statistical interpretation, which leads to a slight under-exclusion compared to the official ATLAS result, which combines the $2\ell$ and $3\ell$ SRs of the same type. The observed limit is about $1\sigma$ weaker than the expected one in the low mass region, and about $1\sigma$ stronger for LSP masses around 300 GeV.

The SRs not being orthogonal, ATLAS-SUSY-2016-24 and ATLAS-SUSY-2017-03 are not combinable in our study; only one of them can enter the global combination for any given parameter point.

**ATLAS-SUSY-2018-05 [37]:** A search in final states with an $e^+e^-$ or $\mu^+\mu^-$ pair, jets, and $\not{E}_T$. This and the following ATLAS analyses are based on 139 fb$^{-1}$ of Run 2 data. The analysis is done in two parts, one targeting $pp \to \tilde{\chi}_1^\pm \tilde{\chi}_2^0 \to (W \tilde{\chi}_1^0)(Z \tilde{\chi}_1^0) \to (q\bar{q}' \tilde{\chi}_1^0)(\ell\bar{\ell} \tilde{\chi}_1^0)$, the other targeting pair production of colored SUSY particles (squarks or gluinos) decaying through the next-to-lightest neutralino. Since these two parts are completely distinct, they are implemented in SMODELS with different analysis IDs: ATLAS-SUSY-2018-05-ewk (comprising EMs for 13 SRs) and ATLAS-SUSY-2018-05-strong (comprising EMs for 30 SRs). The analysis provides extensive material on HEPDATA [52], including acceptance and efficiency values for all SRs, and the full HISTFACTORY statistical models in JSON format [53].

The EWK implementation in SMODELS contains EMs for the $WZ^{(*)} + \not{E}_T$ signature in 13 SRs. We note that these EMs were extracted from the `ewk_signal_patchset.json` file contained in [53], instead of the HEPDATA tables. As can be seen in Figure 6 (top row), the combination of SRs is important to reproduce the ATLAS limit. The full and SIMPLIFY'ed[4] statistical models give very similar results, so the latter is used by default. Control regions are ignored, as including them does not improve the agreement with the official limits from ATLAS. The observed limit is about $1\sigma$ stronger than the expected one.

The SRs of this analysis overlap with those of ATLAS-SUSY-2016-24 and ATLAS-SUSY-2017-03.

**ATLAS-SUSY-2018-06 [38]:** This is an EW-ino search in the three-leptons plus $\not{E}_T$ final state by means of the recursive jigsaw reconstruction technique [54, 55]. It specifically targets the channel $pp \to \tilde{\chi}_1^\pm \tilde{\chi}_2^0 \to (W \tilde{\chi}_1^0)(Z \tilde{\chi}_1^0) \to (\ell \nu \tilde{\chi}_1^0)(\ell\ell \tilde{\chi}_1^0)$. The analysis has two SRs, one vetoing jets, and one requiring 1–3 jets from initial-state radiation. The auxiliary material provided on HEPDATA [56] includes EMs for the $WZ^{(*)} + \not{E}_T$ signature in the 2 SRs as well as the full HISTFACTORY statistical model. A complication arises from the fact that the `BkgOnly.json` model [57] leads to issues[5] which so far could not be clarified. While its SIMPLIFY'ed version works and yields reasonable results, the best SR is also a good approximation and much faster,

---

[4]Produced from the full HISTFACTORY model by means of the SIMPLIFY tool, allowing for a much faster evaluation of the likelihood.

[5]See PYHF issue #1320 for details.

so we stick with using only the best SR for this analysis. The observed limit is about $1\sigma$ weaker than the expected one.

The SRs of this analysis overlap with those of ATLAS-SUSY-2016-24 and ATLAS-SUSY-2017-03.

**ATLAS-SUSY-2018-32 [39]:** A search for electroweak production of charginos or sleptons decaying into final states with two leptons and $\not{E}_T$. It considers events with 0–1 jets, but vetoes $b$-tagged jets. The analysis is made of 36 exclusive SRs (binned in $m_{T2}$) and 16 inclusive SRs (overlapping in $m_{T2}$). The HEPDATA entry [58] includes acceptance and efficiency values of the $pp \to \tilde{\chi}^+\tilde{\chi}^- \to W^+W^- + \not{E}_T$ simplified model for all SRs, as well as the full HISTFACTORY statistical models (`bkgonly.json` and signal patchsets).

Implemented in SMODELS are $WW + \not{E}_T$ EMs for the 36 binned SRs. These were extracted from the `C1C1WW` signal patchsets contained in [59]. The SIMPLIFY'ed statistical model reproduces well the official observed limit but somewhat overestimates the expected limit, see Figure 6 (middle). It is nonetheless used in order to save CPU time. The best SR does not give a sensible limit. We note that this analysis relies on a combined fit of signal and control regions, so the latter must not be pruned when patching the signal counts in the SRs (this means setting `includeCRs:True` in the analysis' `globalInfo.txt` file, see [60]).

It is noteworthy that the SRs of this analysis do not overlap with the SRs of ATLAS-SUSY-2018-05. Indeed, this analysis requires at most 1 jet, while ATLAS-SUSY-2018-05 requires at least 2 jets, except for 1 SR which requires only 1 jet. However, this SR looks for opposite-sign same-flavor (OSSF) leptons with $m_{\ell\ell} \in [71, 111]$ GeV, while ATLAS-SUSY-2018-32 requires the dilepton mass of the OSSF lepton pair to be higher than 121.2 GeV. Moreover, ATLAS-SUSY-2016-24 and ATLAS-SUSY-2017-03 SRs targeting direct EW-ino decays into the LSP with 2 leptons require at least 2 jets. They hence do not overlap with those of ATLAS-SUSY-2018-32.

**ATLAS-SUSY-2018-41 [27]:** A search for charginos and neutralinos in fully hadronic final states, using large-radius jets and jet substructure information to identify high-$p_T$ $W$, $Z$ or Higgs bosons. Two orthogonal SR categories, 4Q and 2B2Q, are defined according to the $qqqq$ and $bbqq$ final states, respectively; events with leptons are vetoed. Multiple SRs are defined in each category to target final states from different combinations of SM bosons ($WW$, $WZ$, $Wh$, $ZZ$, $Zh$, $hh$). Moreover, 3 SRs are inclusive in $V = W, Z$: SR-2B2Q-VZ, SR-2B2Q-Vh and SR-4Q-VV. Acceptance and efficiency tables are provided on HEPDATA [61] for the three inclusive $V = W, Z$ SRs for different signal hypotheses; their implementation in SMODELS covers $WW$, $WZ$, $Wh$, $ZZ$, $Zh$, $hh$ ($+\not{E}_T$) from chargino/neutralino production, provided the masses of the produced particles are not too different.

Since the SR-2B2Q-VZ, SR-2B2Q-Vh and SR-4Q-VV signal regions are described as statistically independent in [27], we combine them by means of a trivial (diagonal) correlation matrix. This means the covariance matrix is given by the background uncertainty squared on the diagonal, $(\text{cov})_{ii} = (\delta b_i)^2$, and zero otherwise.[6] An example of the validation is shown in Figure 6 (bottom). For most scenarios (with the exception of the $WW + \not{E}_T$ channel) the analysis poses a stronger limit than expected. The effect is only about $1\sigma$ but, as we will see, has a strong influence in the combination.

Since leptons are vetoed, the SRs of this analysis do not overlap with any of the other analyses.

**ATLAS-SUSY-2019-02 [40]:** Another search in final states with two leptons plus $\not{E}_T$. Like ATLAS-SUSY-2018-32 above, it considers events with opposite-sign di-leptons (of same or dif-

---

[6]In the meanwhile, ATLAS provided also the full statistical model and extensive patchsets for all SRs on HEP-DATA; their implementation and validation in SMODELS is ongoing.

**Figure 6:** Examples for the validations of the ATLAS-SUSY-2018-05 (top), ATLAS-SUSY-2018-32 (middle) and ATLAS-SUSY-2018-41 (bottom) analyses in SMODELS v2.3.

ferent flavor) with 0 or 1 non-$b$-tagged jets. The difference is that ATLAS-SUSY-2019-02 specifically targets the kinematic region where $m_{\text{mother}} - m_{\text{LSP}}$ is close to the $W$-boson mass. Specific SRs are defined for targeting slepton or chargino production.

Acceptance and efficiency tables are provided on HEPDATA [62] for all SRs and implemented in the SMODELS database. However, no statistical model is available. For the slepton simplified model, the best SR provides a fairly good sensitivity. This is, however, not the case for the chargino simplified model: here, SR combination is essential, as the best SR alone has no sensitivity. Lacking more information, we tried, among others, the assumption that the exclusive different-flavor (DF) SRs are correlated and the exclusive same-flavor (SF) SRs are correlated, but DF and SF are not correlated with each other. This indeed allowed us to

reproduce the official ATLAS limit on $pp \to \tilde{\chi}_1^+ \tilde{\chi}_1^- \to W^+ W^- + \not{E}_T$. Observed and expected limits agree in this case.

The analysis overlaps with ATLAS-SUSY-2018-32, but, since the SRs targeting EW-ino production veto jets, they do not overlap with SRs of other analyses.

**ATLAS-SUSY-2019-08 [41]:** A search for chargino-neutralino pairs in final states with $W \to \ell\nu$ and $h \to b\bar{b}$. The signal selection thus requires one lepton, a pair of $b$-tagged jets consistent with the decay of a Higgs boson, plus $\not{E}_T$. Three sets of SRs target 'low mass', 'medium mass' and 'high mass' scenarios through cuts on the transverse mass variable $m_T$. Each of the three $m_T$ regions is further binned in three $m_{CT}$ regions, resulting in 9 exclusive SRs. In addition, there are 3 inclusive SRs with only lower cuts on $m_T$ and $m_{CT}$.

The analysis provides acceptance and efficiency values for the $Wh + \not{E}_T$ simplified model for all 9 exclusive SRs on HEPDATA [63], together with the full HISTFACTORY statistical model. Combining the 9 SRs by means of the SIMPLIFY'ed statistical model results in a small over-exclusion, so we use the full one in SMODELS, although it requires significantly more CPU time (several minutes instead of $\mathcal{O}(1)$ sec). The validation is shown in Figure 7 (top). The observed limit from this analysis is about $1\sigma$ lower than the expected limit.

Given the $1\ell + 2b$ requirement, the SRs of this analysis do not overlap with any other analysis.

**ATLAS-SUSY-2019-09 [42]:** Another search for chargino-neutralino pairs in three-lepton final states with $\not{E}_T$. Events are classified by jet multiplicity (0 or $\geq 1$ jets) with a veto on $b$-tagged jets. The analysis considers leptonically decaying $W$, $Z$ and SM Higgs bosons with SRs optimised for on-shell $WZ$, off-shell $WZ$ or $Wh$ selections. In total there are 20 SRs for on-shell $WZ$, 31 SRs for off-shell $WZ$ and 21 SRs for $Wh$ selections.

The HEPDATA record [64] provides the full HISTFACTORY statistical models together with patches for the on- and off-shell $WZ$ signal models considered in the paper.[7] Truth-level acceptances and reconstruction efficiencies are also provided on HEPDATA [64] but only for inclusive SRs, which do not allow one to reproduce the official limits from ATLAS. The EMs implemented in SMODELS have therefore been extracted from the JSON patchsets. By default, we use the SIMPLIFY'ed statistical model, which gives almost identical results to the full one, see Figure 7 (bottom). As for ATLAS-SUSY-2018-32, it is important to include the background yields in the control regions in the statistical evaluation.

For the $Wh$ selection, however, no JSON patches are available; this part of the analysis could not be validated and is therefore not included in the SMODELS database. This is a pity, as a small excess (between $1$–$2\sigma$) was observed in this case. A small excess was also reported in the off-shell $WZ$ channel for compressed spectra.

The SRs overlap with ATLAS-SUSY-2016-24, ATLAS-SUSY-2017-03 and ATLAS-SUSY-2018-06.

## 3.2 CMS results

**CMS-SUS-13-012 [43]:** An inclusive search for new physics in multijet events with large $\not{E}_T$ from Run 1. The analysis was designed as a generic search for gluinos and squarks and has 36 SRs characterised by jet multiplicity, the scalar sum of jet transverse momenta, and the $\not{E}_T$. Besides some EMs available on the analysis wiki page, the SMODELS database contains a large number of EMs maps which were obtained through a MADANALYSIS 5 recast [67]; these

---

[7] We note that [65] also includes recipes of how to do a combined fit of the on-shell and the off-shell channels, and how to combine this analysis with the "2-lepton compressed" analysis [66].

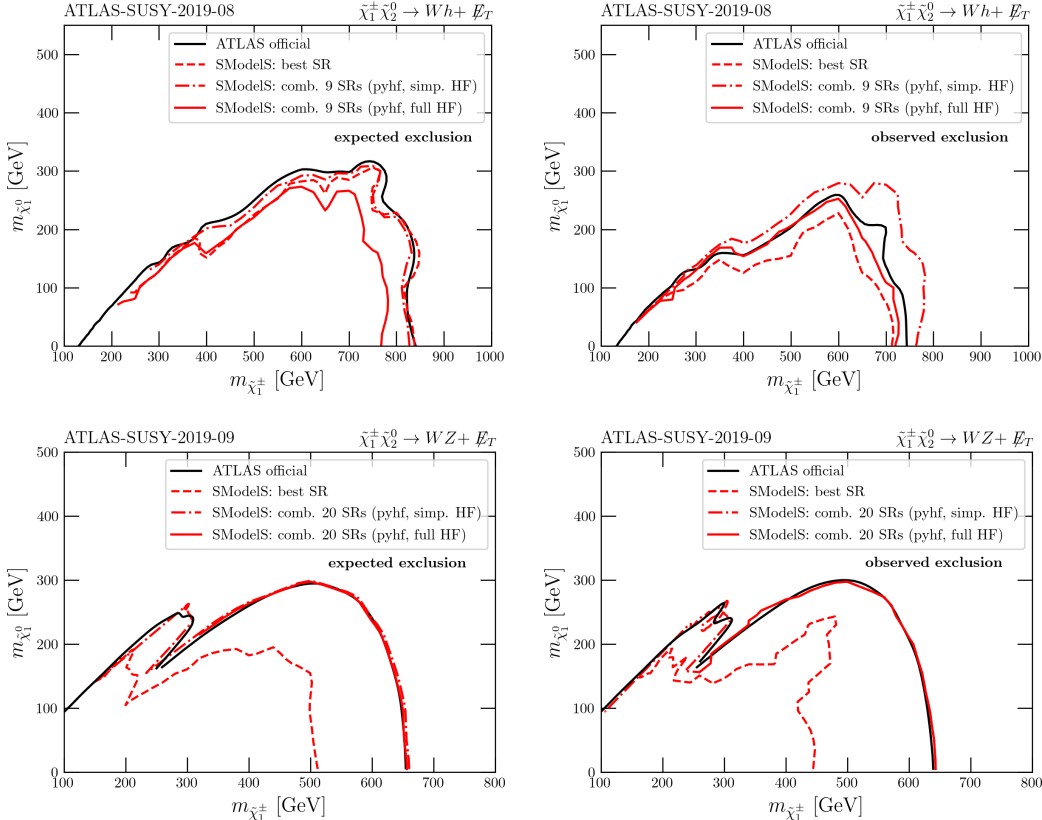

Figure 7: Validation plots for the ATLAS-SUSY-2019-08 and ATLAS-SUSY-2019-09 analyses in SMODELS v2.3.

include also $WW$, $WZ$ and $ZZ$ EMs for chargino/neutralino production. Only the best SR is used for limit setting.

**CMS-SUS-16-039** [44]: A search for charginos and neutralinos in multilepton final states from Run 2 with 36 fb$^{-1}$. Considered are final states with 2 same-sign leptons as well as final states with 3–4 leptons including up to 2 $\tau$'s. The analysis is made of 11 search categories, each subdivided into up to 44 SRs. CMS provided EMs for a set of 8 "super signal regions" defined by simpler selections, but these turned out to have very little sensitivity. The EMs used in SMODELS were therefore obtained through a MADANALYSIS 5 recast [68]. These are for the "category A" SRs (3 $e/\mu$'s that form at least one OSSF pair). To save CPU time, the 43 SRs of this category are aggregated to 11 in the SMODELS database. The covariance matrix provided by CMS is used to combine the SRs. The combination of the 11 aggregated SRs reproduces fairly well the observed limit from CMS but underestimates the expected exclusion, see Figure 8 (top). We also note that the analysis saw a slight excess in $3\ell + \not{E}_T$ events, so the observed limit is slightly weaker than expected.

**CMS-SUS-16-048** [45]: This is a 36 fb$^{-1}$ Run 2 search in final states with two soft, oppositely charged leptons of same or different flavor, originating from off-shell $W$ and $Z$ boson decays, and $\not{E}_T$. To reduce the SM background, the 2 leptons are required to be produced along with at least one hard, non-$b$-tagged jet, which is supposed to come from ISR. The search is then sensitive to lepton transverse momenta between 5 and 30 GeV.

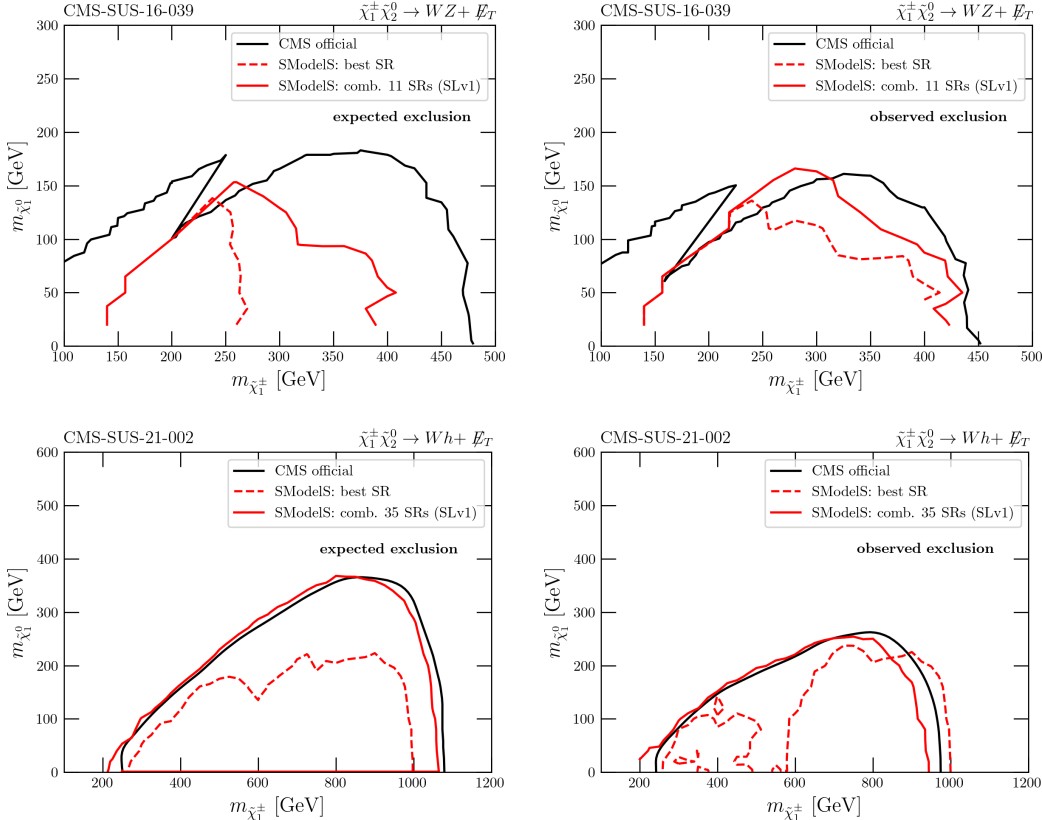

Figure 8: Validation plots for the CMS-SUS-16-039 and CMS-SUS-21-002 analyses in SMODELS v2.3.

This way the analysis targets compressed spectra of EW-inos or stops, with 12 SRs optimised for $\tilde{\chi}_1^\pm \tilde{\chi}_2^0$ and 9 SRs optimised for stops. CMS supplied a covariance matrix for the 21 SRs; the corresponding efficiency×acceptance values for EW-inos or stops were, however, not made available. The EMs implemented in the SMODELS database were therefore obtained through a MADANALYSIS 5 recast [69]. For mass differences around 20 GeV, the observed limit is about $1\sigma$ stronger than the expected one.

The CMS-SUS-16-048 analysis and CMS-SUS-16-039 are orthogonal to each other.

**CMS-SUS-20-004** [46]: This is a search for new physics in channels with two Higgs bosons, each decaying into $b\bar{b}$, and large $\not{E}_T$, using 137 fb$^{-1}$ of Run 2 data. Events with leptons are vetoed. It comprises 22 SRs with 3–4 $b$-jets, which are binned in $p_T$; 16 of these SRs target a resolved signature and 6 target a boosted signature.

CMS provides a covariance matrix for the 22 SRs together with EMs for a $pp \to \tilde{\chi}_2^0 \tilde{\chi}_2^0 \to (h\tilde{\chi}_1^0)(h\tilde{\chi}_1^0)$ simplified model on HEPDATA [70], which we implemented in the SMODELS database. Since the reported uncertainties are asymmetric, the SLv2 (Gaussian with a skew) [14, 47] approach is used for SR combination. The necessity of going beyond the Gaussian approximation for this analysis was discussed in detail in [71]. We note that this analysis sees deviations from the SM of the level of about 1–2$\sigma$ in several SRs.

Given the multi $b$ requirement, the SRs of this analysis do not overlap with any other analysis.

**CMS-SUS-21-002 [28]:** This is the CMS all-hadronic search for electroweak-inos with 137 fb$^{-1}$ of Run 2 data. It considers final states with large $\not{E}_T$ and pairs of hadronically decaying SM bosons ($W$, $Z$ or Higgs), which are identified using novel algorithms. In total 35 SRs are defined in one $b$-veto search region (0 $b$-tagged jets) and three $b$-tag search regions ($\geq 1$ $b$-tagged jets). Events with leptons are vetoed.

Efficiency maps for $Wh$, $WZ$ and $WW + \not{E}_T$ simplified models as well as a covariance matrix for all SRs are available as ROOT files on the analysis wiki page (but not on HEPDATA). A validation example illustrating the importance of the SR combination is shown in Figure 8 (bottom). The observed limit is weaker than the expected one, which signals an excess in observed events. Like for the hadronic ATLAS search (which saw a deficit), the effect is of the order of $1\sigma$ but given the high sensitivity of the analysis it will play an important role in the global likelihood.

CMS-SUS-20-004 and CMS-SUS-21-002 overlap but are orthogonal to the other CMS analyses.

## 4 Setup of the numerical analysis

### 4.1 Parameter scan

As already done in [14], we reuse the EW-ino scan points from [13] for our numerical analysis. This not only saves CPU time, it also allows for direct comparison with the earlier publications. In section 4.2 of [13], $M_1$, $M_2$, $\mu$, and $\tan\beta$ were randomly scanned over within the following ranges:

$$
\begin{aligned}
10 \text{ GeV} < \quad & M_1 \quad < 3 \text{ TeV}, \\
100 \text{ GeV} < \quad & M_2 \quad < 3 \text{ TeV}, \\
100 \text{ GeV} < \quad & \mu \quad < 3 \text{ TeV}, \\
5 < \quad & \tan\beta \quad < 50.
\end{aligned}
$$

All other SUSY breaking parameters were fixed to 10 TeV, assuming that the stop-sector parameters can always be adjusted such that $m_h \simeq 125$ GeV without influencing the EW-ino sector. The lower limits on $M_2$ and $\mu$ were chosen so as to avoid the LEP constraints on light charginos, while the bounds on $\tan\beta$ were chosen to avoid Yukawa couplings from becoming non-perturbatively large. The mass spectra and decay tables were computed with SOFT-SUSY 4.1.11, setting $m_h = 125$ GeV for consistency of the decay calculations. All points have a neutralino LSP.

From the close to 100k points of the complete scan in [13], we here select the subset of points with only prompt decays (no long-lived particles, all decay widths $\Gamma_{tot} > 10^{-11}$ GeV). Moreover, we require $m_{\tilde{\chi}_1^0} < 500$ GeV and $m_{\tilde{\chi}_1^\pm} < 1200$ GeV in order to focus on the region which the current prompt EW-ino searches are sensitive to. Removing a few points with $m_{\tilde{\chi}_1^\pm} < 103$ GeV as well as a few points with erroneous branching ratios in SOFTSUSY 4.1.11[8] leaves us with a total of 18247 points as the EW-ino dataset for this study. An important update with respect to [13, 14] is that the production cross sections have now been computed at next-to-leading order (NLO) with RESUMMINO 3.1.2 [19, 20] if the corresponding LO cross section was above $7.10^{-4}$ fb. The interface to RESUMMINO is described in Appendix A.

In what follows, it is often relevant to distinguish between scenarios with a bino-like LSP and those with non-bino-like LSP. As bino-like LSP we consider a $\tilde{\chi}_1^0$ with at least 50% bino

---

[8]Due to a bug in the transition from 2-body to 3-body neutralino decays, which was in the meanwhile corrected, see https://ballanach.github.io/softsusy/.

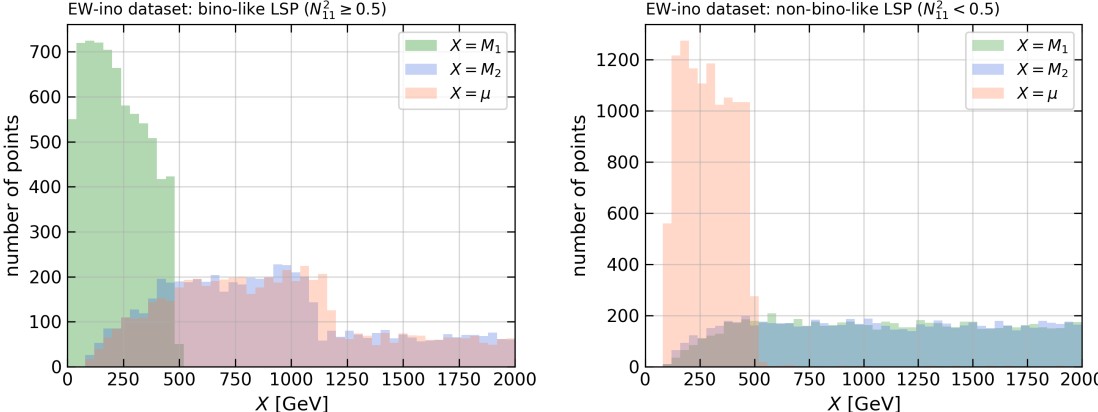

Figure 9: Distributions of $M_1$, $M_2$ and $\mu$ in the set of EW-ino scan points used in this study; on the left for points with a bino-like LSP, on the right for points with a non-bino-like LSP. Only values up to 2 TeV are shown.

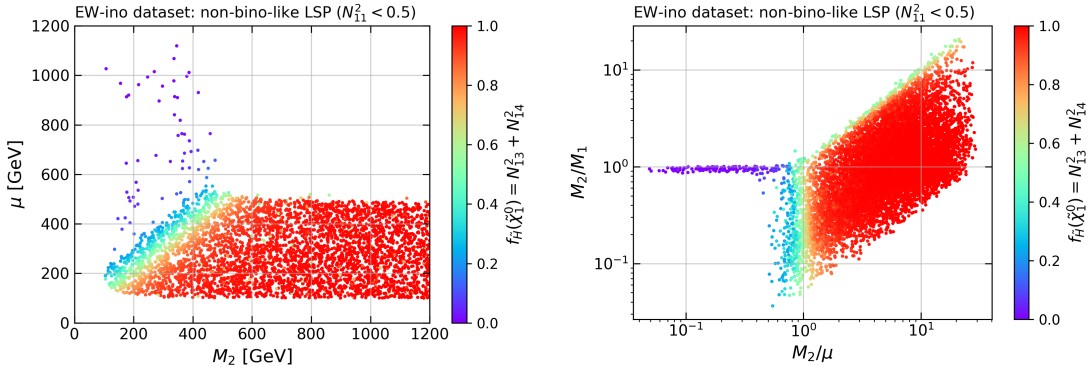

Figure 10: Non-bino-like LSP points, on the left in the $\mu$ vs. $M_2$ plane and on the right in the $M_2/M_1$ vs. $M_2/\mu$ plane. The colour code shows higgsino content of the LSP. In the left plot, the $x$- and $y$-axes go only up to 1.2 TeV for better visibility of the mixed region, but in both directions scan points extend up to 3 TeV.

component, i.e. $N_{11}^2 \geq 0.5$, where $N$ is the neutralino mixing matrix defined in eq. (4). In turn, non-bino-like LSP points ($N_{11}^2 < 0.5$) feature a mostly higgsino-like, mostly wino-like, or strongly mixed $\tilde{\chi}_1^0$. Figure 9 shows the distributions of $M_1$, $M_2$ and $\mu$ values in our dataset; in the left plot for points with a bino-like LSP and in the right plot for points with a non-bino-like LSP. The cutoff at $M_1 \approx 500$ GeV in the plot on the left comes from the requirement $m_{\tilde{\chi}_1^0} < 500$ GeV, while the edges around $M_2$, $\mu \approx 1.2$ TeV come from $m_{\tilde{\chi}_1^\pm} < 1200$ GeV. In the plot on the right, $m_{\tilde{\chi}_1^0} < 500$ GeV together with the requirement of promptly decaying $\tilde{\chi}_1^\pm$ leads to the edge at $\mu \approx 500$ GeV (higher values of $\mu$ being hardly visible in the plot), while $M_1$ and $M_2$ can range from small to very large values. That the $M_2$ distribution is suppressed towards small values, instead of showing an edge around 500 GeV, is due to the fact that light winos are generally long-lived; as mentioned we removed these cases from the scan. Also, there is no edge around 1.2 TeV, because for non-bino-like LSP points the $\tilde{\chi}_1^\pm$ is always close in mass to the $\tilde{\chi}_1^0$ and thus $m_{\tilde{\chi}_1^\pm} < 540$ GeV.

The properties of non-bino-like LSP points can be understood from Figure 10, which shows these points in the $\mu$ vs. $M_2$ plane (left) and the $M_2/M_1$ vs. $M_2/\mu$ plane (right). The colour code indicates the higgsino content of the LSP: for red points the $\tilde{\chi}_1^0$ is mostly higgsino, while

for purple points it is mostly wino. In between, from yellow to green to turquoise, the $\tilde{\chi}_1^0$ is a strongly mixed state of higgsino and wino ($\mu \approx M_2$), higgsino and bino ($\mu \approx M_1$) or wino and bino ($M_2 \approx M_1$). The sparsity of points at low $M_2$ (visible in the left plot) is due to the lifetime constraint, as points with long-lived charginos have been removed from the scan. In fact, for all the remaining points in the region $M_2 < 500$ GeV and $\mu \gtrsim M_2 + 100$ GeV in Figure 10, the bino mass parameter $M_1$ is close to $M_2$, roughly $M_1 \in [1, 1.2] \times M_2$. This increases the splitting between the $\tilde{\chi}_1^0$ and the $\tilde{\chi}_1^\pm$, such that the latter is no longer long-lived.

Each point in the EW-ino dataset is confronted against the experimental results of the analyses detailed in Section 3. To this end, SMODELS computes the likelihood $\mathcal{L}(\mu, \theta | D)$ for the respective SUSY signal for each analysis. This likelihood describes the plausibility of a signal strength $\mu$ given the data $D$, with $\theta$ denoting the nuisance parameters, see [14] for details. Given the likelihood, a 95% confidence level limit on the signal strength, $\mu_{95}$, is computed using the $\text{CL}_s$ prescription [72]. The result is reported in the form of $r$-values, with $r$ defined as the ratio of the predicted fiducial cross section of the signal over the corresponding upper limit ($r \equiv 1/\mu_{95}$). The standard SMODELS output consists of the expected and observed $r$-values, $r_{\text{exp}}$ and $r_{\text{obs}}$, for each analysis. Often, a point is considered excluded if the highest observed $r$-value $r_{\text{obs}}^{\max} \geq 1$. This corresponds to an exclusion by the *most constraining* analysis. The statistically more sound approach, however, is to base the exclusion on the *most sensitive* (or "best") analysis, i.e. the analysis with the highest $r_{\text{exp}}$; a point is then considered as excluded if the corresponding observed $r$-value $r_{\text{obs}}^{\text{best}} \geq 1$. As discussed also in [14], either approach is quite sensitive to statistical fluctuations, which motivates us to attempt a global combination.

When running SMODELS, we use a `sigmacut` of $10^{-3}$ fb; this parameter sets the minimum cross section value considered in SLHA decomposition. Moreover, we use a `minmassgap` value of 10 GeV (which differs from the default value of 5 GeV). This parameter defines the minimum mass difference for mass compression in SMODELS:[9] If BSM particle $P_2$ decays into BSM particle $P_1$ but the mass difference $m_{P_2} - m_{P_1} < $ `minmassgap`, the decay products are assumed to be potentially too soft to be visible in a typical LHC analysis. Two signal topologies are hence compared with the simplified model results in the database: the compressed one, where the decay $P_2 \rightarrow P_1 + X$, $X$ being any SM decay product(s), is ignored, and the non-compressed one, where the decay is kept. This is relevant in particular if one or more SUSY particles are very close in mass to the LSP, as is the case for non-bino-like LSP points. Our choice of `minmassgap` and its influence on the results is explained in Appendix B.

## 4.2 Combination strategy

The next step is to build a global likelihood from the individual analysis likelihoods. Since we do not have access to inter-analyses correlations, only analyses that are approximately uncorrelated may enter the combination. The combined likelihood $\mathcal{L}_C$ is then simply the product of the likelihoods $\mathcal{L}_i$ of the individual analyses. This raises two questions: i) which analyses can be treated as uncorrelated, and ii) how to choose the best combination from all possible combinations.

Regarding i), we assume analyses from different LHC runs, and from distinct experiments (ATLAS or CMS) to be uncorrelated. Furthermore, we also treat analyses which do not share any event in their SRs as approximately uncorrelated. In [15], the latter aspect was limited to clearly different final states (e.g., fully hadronic final states vs. final states with leptons). In this work, we go a step further and carefully scrutinise the SR definitions of all analyses under consideration (see the descriptions in Section 3) to determine whether they are orthogonal or they overlap. With this procedure, we arrive at the combinability matrix shown in Figure 11.

---

[9]A detailed description of the mass compression procedure is available in the SMODELS online documentation: https://smodels.readthedocs.io/en/latest/Decomposition.html#masscomp

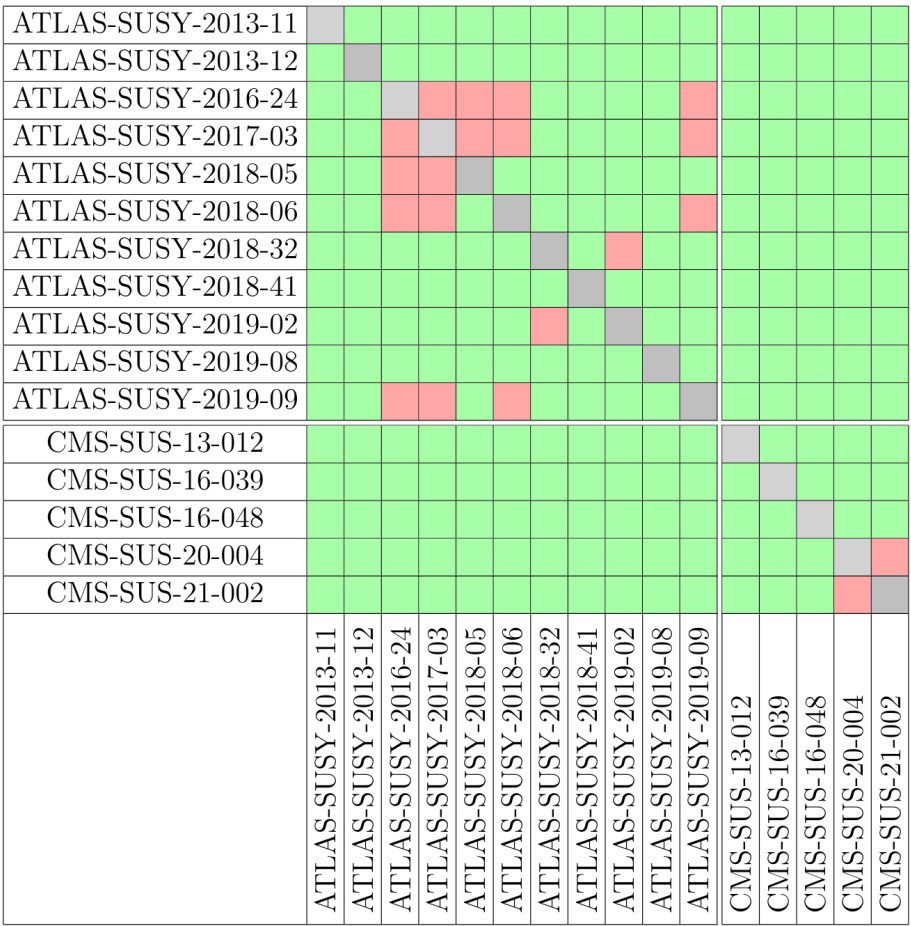

Figure 11: Matrix displaying the combinability of the searches considered in this study. Green means the two analyses are assumed to be approximately uncorrelated and can be combined, while red means that they are not and shouldn't be combined.

It should be noted that our approximation assumes that inter-analyses correlations of systematic uncertainties (stemming from, e.g., luminosity measurements) can be neglected. Moreover, it neglects possible correlations of systematic uncertainties due to overlaps of SRs of one analysis with the control regions of another analysis. Such effects can only be checked in full recasts including signal *and* control regions, which is beyond the present state-of-the-art. We do, however, expect that the effects are small compared to other uncertainties in SMODELS and in reinterpretation studies in general.

With respect to ii), since not all the analyses can enter the same combination, many different combinations can be built based on this combinability matrix. In this work, the "best combination" is defined as the most sensitive one, i.e. the one with the lowest expected limit on the signal strength $\mu_{95}^{\exp}$ (or, equivalently, highest $r_{\exp}$). Technically, since the computation of $\mu_{95}^{\exp}$ can be time-consuming, we assume that the combination that minimizes

$$w = \frac{\mathcal{L}^{\exp}(\text{BSM})}{\mathcal{L}^{\exp}(\text{SM})}, \tag{13}$$

is the one with the lowest $\mu_{95}^{\exp}$. Here, $\mathcal{L}^{\exp}(\text{H})$ is the expected combined likelihood under the H hypothesis. For a given model point, the series of steps to find the best combination of analyses is the following:

1. Compute $\mu_{95}^{\exp}$ for every analysis and keep for the following steps only the sensitive ones.

By sensitive we here mean $\mu_{95}^{\text{exp}} \leq 10$. Even though the insensitive analyses could have been used in the combination, they do not impact the result and hence were removed for technical stability and reduction of the computation time.

2. Identify all potential combinations according to the combinability matrix, Figure 11. If there is only one analysis sensitive to the tested model, the "combination" corresponds to that individual analysis.

3. Remove all the combinations that are subsets of other combinations. This ensures that all the available information is used to build the combination.

4. Compute $w$ for all the remaining combinations and select the one that gives the lowest value.

In practice, the combiner algorithm from [15] was used with minimal modifications to determine the best combination for each point in our EW-ino dataset. We cross-checked the procedure with a second method, based on the "pathfinder" algorithm from [73], which we adapted for our study to find the best combination of analyses, instead of the optimal SR combination in [73]. Both methods gave identical results.

In the following, we will refer to the best combination of analyses simply as "the combination".

# 5 Results

Using the parameter scan and combination strategy discussed in the previous section, we investigate the impact of analysis combination on the EW-ino sector of the MSSM. The gain in expected reach due to the combination can be seen in the left plot of Figure 12. A point is expected to be excluded if an analysis gives $r_{\text{exp}} \geq 1$. The expected exclusion is mostly driven by three analyses: the $3\ell + \slashed{E}_T$ search ATLAS-SUSY-2019-09 and the all-hadronic searches CMS-SUS-21-002 and ATLAS-SUSY-2018-41. The points for which one of these analyses is the most sensitive one are displayed by the orange, light blue and light red histograms. As anticipated, the expected reach is enhanced due to the accumulation of statistics within the combination, resulting in an increase of 48% in the expected number of excluded points (from 3081 to 4549 points).

The impact of the combination on the observed number of excluded points with respect to the most sensitive and most constraining analyses, i.e. the analyses that give the highest $r_{\text{exp}}$ and $r_{\text{obs}}$ respectively, is shown on the right plot of Figure 12. We point out that using the largest $r_{\text{obs}}$ is not statistically sound, since it gives preference to the analyses which have observed under-fluctuations in the data, thus artificially increasing the constraining power. The histogram for the most constraining analyses in Figure 12 is shown only for reference and comparison to previous studies. Nonetheless, it is interesting to note that the combination excludes more points than the most constraining analysis up to $m_{\tilde{\chi}_1^{\pm}} \approx 700$ GeV. For higher masses, the under-fluctuations recorded by the hadronic ATLAS analysis result in a more aggressive exclusion when considering the most constraining analysis. All in all, the most sensitive analysis, the most constraining analysis and the combination exclude 3046, 3949 and 4124 points, respectively. Regarding the importance of using NLO cross sections, 611 points ($\approx 15\%$) would not have been excluded by the combination had we used LO instead of NLO cross sections.

It is also relevant to verify which analyses are entering the combination as we move around the parameter space. In particular, it is interesting to check the stability of the combinations and the number of analyses contributing to each combination. We first show in Figure 13 the combinations for the excluded points featuring a bino-like LSP, in the plane of $m_{\tilde{\chi}_1^0}$ vs. $m_{\tilde{\chi}_1^{\pm}}$.

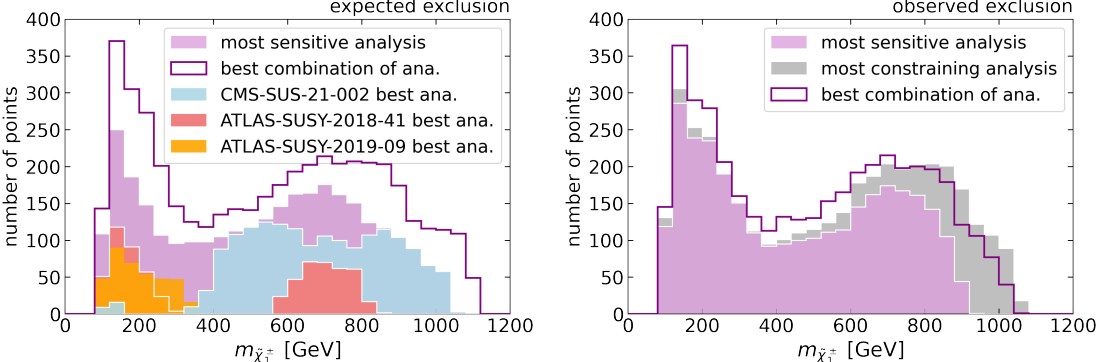

Figure 12: Number of points expected to be excluded (left) and number of points actually excluded (right) as determined by the most sensitive analysis (pink area) and the best combination of analyses (purple line). In addition, the left panel shows the impact of the two hadronic searches: CMS-SUS-21-002 (light blue) and ATLAS-SUSY-2018-41 (light red), along with the $3\ell + \not{E}_T$ search ATLAS-SUSY-2019-09 (orange). On the other hand, the right panel also displays the observed exclusion with respect to the most constraining analysis (grey).

Two observations can be made. First, the combinations are clustered in different regions of the parameter space, which proves the stability of the procedure. Second, the higher the mass of $\tilde{\chi}_1^\pm$ or $\tilde{\chi}_1^0$, the fewer analyses enter the combination. This is because the majority of them are sensitive to low masses, while only ATLAS-SUSY-2018-05, ATLAS-SUSY-2018-41, ATLAS-SUSY-2019-08 and CMS-SUS-21-002 are sensitive to high masses. Indeed, moving from one colour patch to another can be understood as an analysis becoming (in)sensitive to the tested signal when $m_{\tilde{\chi}_1^0}$ or $m_{\tilde{\chi}_1^\pm}$ (increases) decreases. The (small) overlapping regions are due to the composition of the produced particles: in the wino-bino scenario the production cross sections are larger and the sensitivity of the analyses are extended, while for the higgsino-bino scenario the sensitivity is suppressed. Finally, the remaining grey points are made of many different combinations and are concentrated at low masses due to a large number of analyses being sensitive only in this region.

Figure 14 displays the same information, but for points with a non-bino-like LSP (i.e. points with $N_{11}^2 < 0.5$). Since points featuring a long-lived $\tilde{\chi}_1^\pm$ were removed, the majority of points have a higgsino-like LSP. The number of points excluded by the combination is highly reduced for these scenarios. Indeed, as discussed in Section 2.2, the higgsino-like LSP scenario produces a large number of final state topologies, many of which have no available LHC results. As already seen in Figure 13, the low mass region is populated by combinations made up of many analyses, while only four analyses are sensitive enough to enter the combination at high masses. The grey points at low LSP masses contain a sufficiently large $m_{\tilde{\chi}_1^\pm} - m_{\tilde{\chi}_1^0}$ mass gap and are excluded by higgsinos ($\tilde{\chi}_1^\pm$ and $\tilde{\chi}_2^0$) decays to off-shell gauge bosons. Finally, the two brown points at large $m_{\tilde{\chi}_2^\pm}$ are constrained by a combination of analyses sensitive to off-shell higgsino decays and on-shell wino decays.

The impact of analysis combination can be better understood once we consider the individual likelihoods for the analyses entering the combination and the combined likelihood. The top plots of Figure 15 display the combination of 8 analyses for a sample point (P1) featuring a bino-like LSP with similar wino and higgsino mass scales. The left plot shows the individual and combined likelihoods, while the right plot shows the evolution of the combined $r$-values, when analyses are sequentially added to the combination. The most sensitive analysis is expected to exclude the point with $r_{\text{exp}} = 1.09$, but due to an over-fluctuation seen by the ATLAS

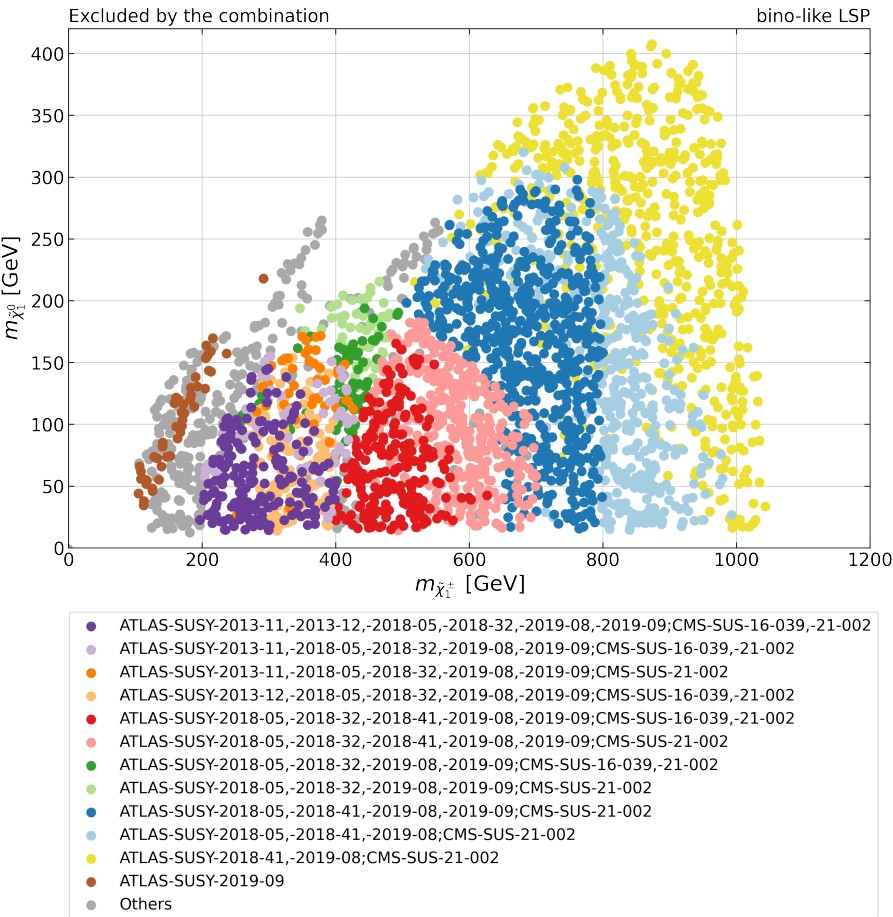

Figure 13: Most sensitive combinations in the $m_{\tilde{\chi}_1^0}$ vs. $m_{\tilde{\chi}_1^\pm}$ plane for points excluded by the combination and featuring a bino-like LSP. Only the combinations appearing at least 25 times are shown. The combinations that appeared less than 25 times fall into the "Others" category.

search, the observed $r$-value is below 1. However, once we combine 8 analyses, the effect of over and under-fluctuations are drastically reduced and the point is robustly excluded with an observed $r$-value very close to the expected value. This can be seen on the left plot, where the analyses that saw a small excess ($r_{\text{obs}} < r_{\text{exp}}$) pull the combined likelihood towards positive values of the signal strength $\mu$, while the analyses that saw an under-fluctuation ($r_{\text{obs}} > r_{\text{exp}}$) pull the combined likelihood in the opposite direction. In this case, the resulting likelihood is centered around 0, i.e. its observed $r$-value is close to its expected one, and its width is narrow, meaning that the observed $r$-value is high. One can also see on the right plot that the small excess recorded by the most sensitive analysis is already compensated by the second most sensitive analysis, which recorded an under-fluctuation, when the combination is only made of these two analyses.

The bottom plots of Figure 15 show the results for a point (P2) representing the wino-bino scenario. This point is neither expected to be excluded nor excluded by any individual analysis entering the combination. However, the constraining power is considerably increased once we combine the 8 analyses shown, excluding the point with $r_{\text{obs}} = 1.42$. It is also interesting to note that, while the combination of the two most sensitive analyses is already sufficient to exclude the point, the inclusion of the remaining analyses significantly increases the sensitivity of the combination.

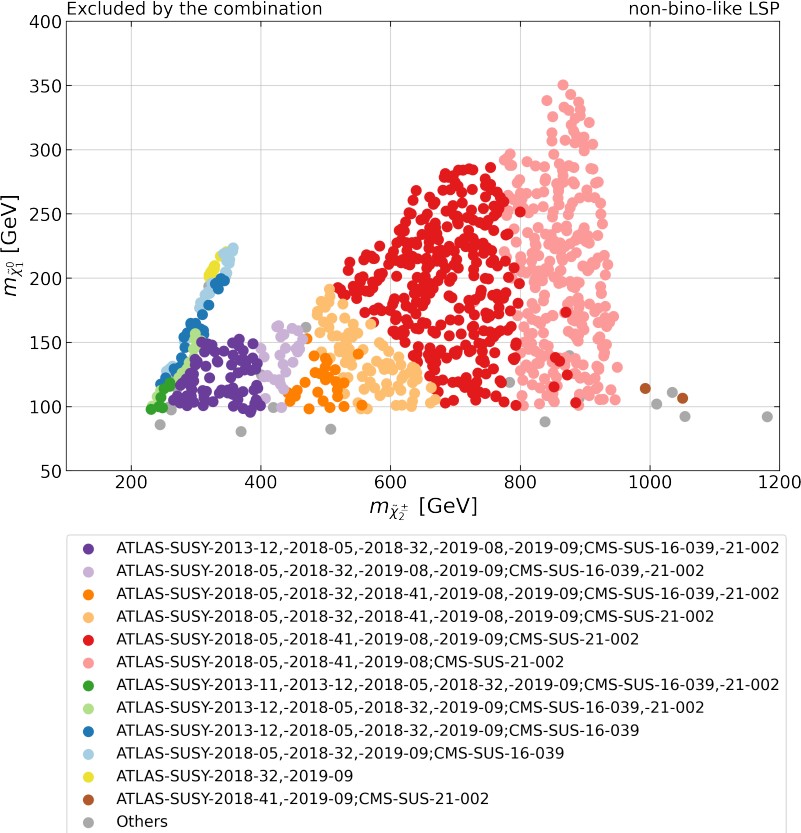

Figure 14: Most sensitive combinations in the $m_{\tilde{\chi}_1^0}$ vs. $m_{\tilde{\chi}_2^\pm}$ plane for points excluded by the combination and featuring a non-bino-like LSP. Only the combinations appearing at least 2 times are shown. The other combinations that appeared 2 times or less fall into the "Others" category.

If the number of sensitive analyses entering the combination is not sufficiently large, it can happen that observed fluctuations are not levelled and the observed $r$-value can be quite distinct from the expected value. This is illustrated by the example in the top plots of Figure 16, which shows the results for a wino-bino point (P3). In this case, the two most sensitive analyses (CMS-SUS-21-002 and ATLAS-SUSY-2019-08) saw an excess, which reduced their observed exclusion. Although the other two analyses entering the combination could mitigate the excess, they are not sufficiently sensitive to have a significant impact on the final $r$-value. As a result, although the combination is expected to exclude the point, $r_{\text{obs}}^{\text{comb}} < 1$.

Finally, there are cases where the combination decreases the *observed* $r$-value when compared to the most sensitive analysis. This can take place when the latter observes an under-fluctuation, thus increasing $r_{\text{obs}}$, while the combination reduces this effect, resulting in a smaller value for $r_{\text{obs}}$. This is illustrated by the bottom plots in Figure 16, which shows the results for a mixed scenario (P4). The most sensitive analysis in this case is the ATLAS-SUSY-2018-41 search, which recorded an under-fluctuation, resulting in an exclusion: $r_{\text{obs}} = 1.11 > 1 > r_{\text{exp}}$. Once we include the CMS-SUS-21-002 search, which saw an excess, the combined $r_{\text{obs}}$ is reduced below one and the point is no longer excluded. This result is strengthened once we include the remaining third analysis entering the combination. It is expected, however, that once additional sensitive analyses enter the combination, such effects will be reduced (if they did not record an excess), and the combined observed exclusion will be quite similar to its expected value.

**P1**



**P2**

Figure 15: Normalised likelihood versus the signal strength $\mu$ (left plots) for 2 different points of the random scan, denoted as P1 (top) and P2 (bottom). Shown are the combined likelihood (black dashed line) together with the likelihoods of every analysis entering the combination (coloured full lines). The smaller panels to the right show the evolution of the combined observed and expected $r$-values when the analyses entering the combination are added one by one in order of decreasing sensitivity. The boxes below the likelihood plots provide further details on the parameter points; $\tilde{\chi}_1^{\pm}$ decays are not specified as the only available decay mode is into $W + \tilde{\chi}_1^0$. Both points are excluded by the combination.

**P3**

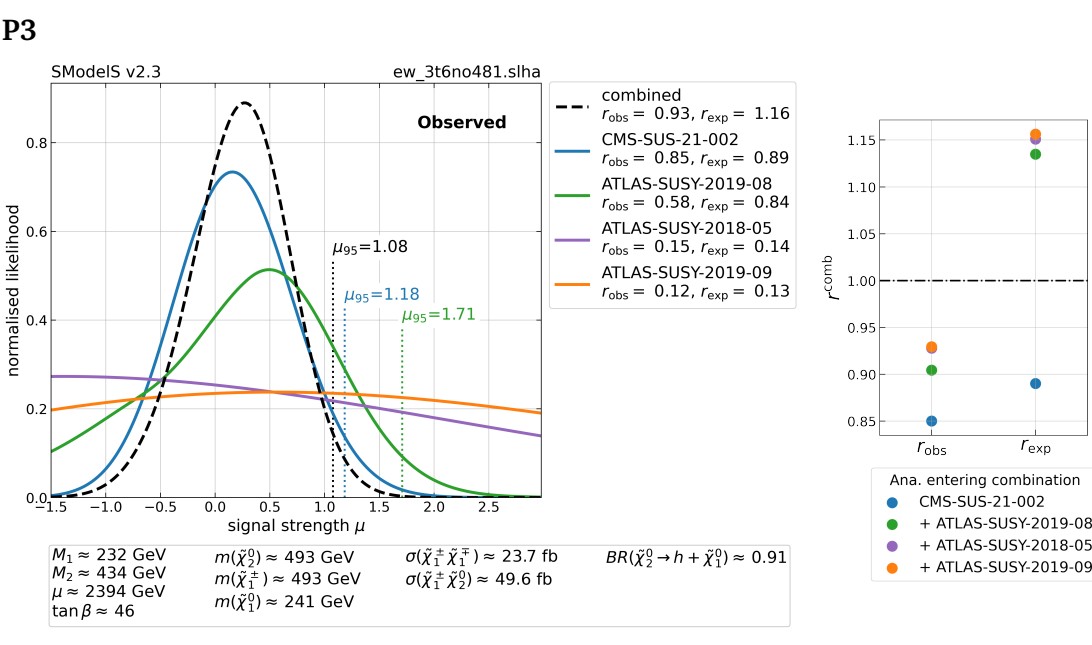

**P4**

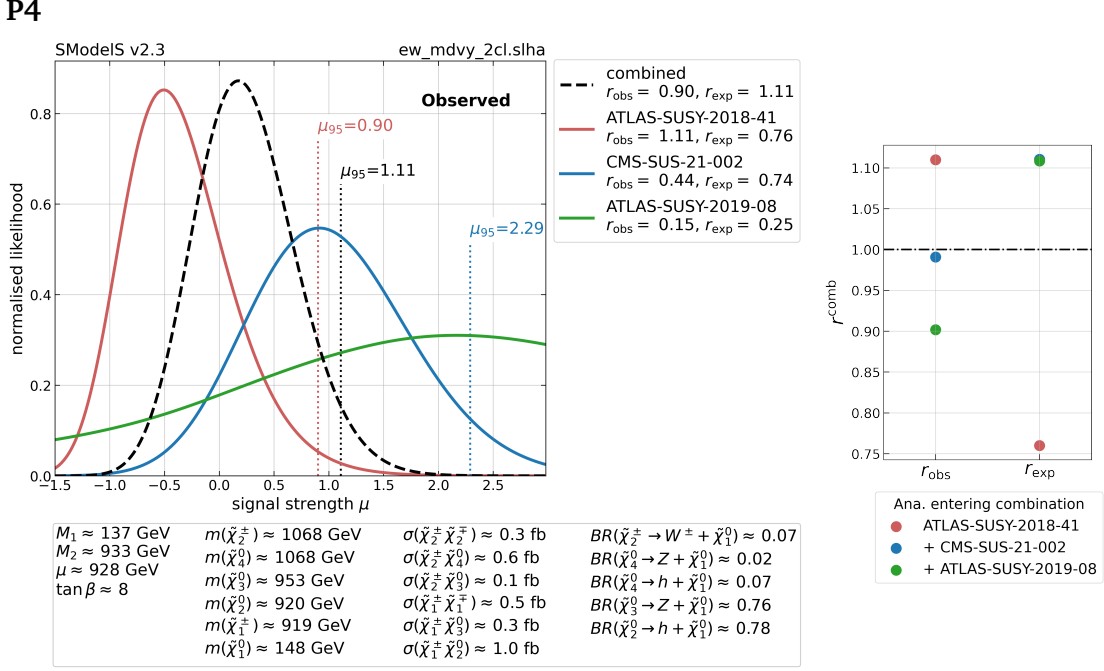

Figure 16: As Figure 15 but for two sample points, P3 (top) and P4 (bottom), not excluded by the combination.

A global overview of the impact of analysis combination is shown in Figure 17 for excluded points with a bino-like LSP. Red (blue) points correspond to a gain (loss) in exclusion power when compared to the most sensitive analysis. The numbered points highlighted in the plot correspond to the points in Figures 15 and 16. The majority of points display an increase in exclusion power, as expected. However, for 700 GeV $\lesssim m_{\tilde{\chi}_1^\pm} \lesssim$ 900 GeV, there is a region where the observed $r$-value for the combination is smaller than the one obtained using only the most sensitive analysis. This is due to the most sensitive analysis (ATLAS-SUSY-2018-41) recording an under-fluctuation, which increased its observed exclusion power. When combined with the other analyses, which recorded an excess, the combined $r_{\text{obs}}$ is reduced, as illustrated by the example shown in Figure 16 (bottom plot). Finally, the bright yellow regions display points for which the impact of combining analysis is negligible. For the points around $m_{\tilde{\chi}_1^\pm} \simeq$ 700 GeV this happens because the analyses entering the combination have observed under and over-fluctuations, thus pulling the combined likelihood in opposite directions. The combined result in this case does not significantly increase the constraining power when compared to the most sensitive analysis. The yellow region at lower masses, $m_{\tilde{\chi}_1^\pm} \lesssim$ 200 GeV, are so highly constrained by the $3\ell + \not{E}_T$ search ATLAS-SUSY-2019-09, that adding new analyses to the combination almost does not change the result.

In the bottom plot of Figure 17 we show only the points which were not excluded by the most sensitive analysis but are now excluded by the combination (orange and red) and the points which were excluded by the most sensitive analysis but are now not excluded by the combination (blue). We can see that the blue points are concentrated in a small region around $m_{\tilde{\chi}_1^\pm} \simeq$ 900 GeV, where red points are also found. All these blue points feature a higgsino-like next-to-LSP (NLSP), while the red points in the same region feature a wino-like NLSP. This behaviour can be explained because the ATLAS hadronic search (ATLAS-SUSY-2018-41) is the analysis the most sensitive to the higgsino-bino scenario in this region and, as already pointed out above, the under-fluctuations recorded by this analysis result in an increase in its constraining power with respect to the combination. On the other hand, in this region, the most sensitive analysis to a wino-bino scenario is the CMS hadronic search (CMS-SUS-21-002), which recorded an excess. A large number of points hence become excluded when it is combined with the ATLAS hadronic search.

Figure 18 shows the same plots but for non-bino-like LSP points in the plane of $m_{\tilde{\chi}_1^0}$ vs. $m_{\tilde{\chi}_2^\pm}$. Most of these feature a higgsino-like LSP and correspond to the wino-higgsino or mixed scenarios. The three most sensitive analyses in this case are the $3\ell + \not{E}_T$ search ATLAS-SUSY-2019-09 for 200 GeV $\lesssim m_{\tilde{\chi}_2^\pm} \lesssim$ 400 GeV, the CMS hadronic search for 400 GeV $\lesssim m_{\tilde{\chi}_2^\pm} \lesssim$ 600 GeV, and the ATLAS hadronic search (ATLAS-SUSY-2018-41) for $m_{\tilde{\chi}_2^\pm}$ between 600 GeV and 1000 GeV. As seen for the bino-like LSP points, the first two mass regions display an increase in constraining power due to analysis combination. Once again, for the high mass region ($m_{\tilde{\chi}_2^\pm} \gtrsim$ 700 GeV), the under-fluctuations recorded by the hadronic ATLAS search result in an increase of constraining power, thus explaining the blue points in Figure 18.

The bottom plot shows only the set of points which have its exclusion status modified by the combination. The low-mass region shows the gain in exclusion power (red points), while the high-mass region display points which become unexcluded by the combination (blue points). Unlike the bino-like LSP scenario, the blue points are approximately clustered into two disjoint regions. This behavior is caused by the `minmassgap` value chosen and the mass compression procedure (see discussion in Appendix B), which compresses topologies with $m_{\tilde{\chi}_2^0} - m_{\tilde{\chi}_1^0} <$ 10 GeV. Only for $m_{\tilde{\chi}_2^\pm} \gtrsim$ 900 GeV this mass difference falls below 10 GeV and some complex topologies generated by two step cascade decays are compressed to single step topologies, thus increasing the sensitivity of the analyses. As a result, even though the analysis combination reduces $r_{\text{obs}}$, its values are still sufficiently large to not alter the exclusion status of the points in this region, resulting in a lack of points around 900 GeV. For larger $m_{\tilde{\chi}_2^\pm}$ the

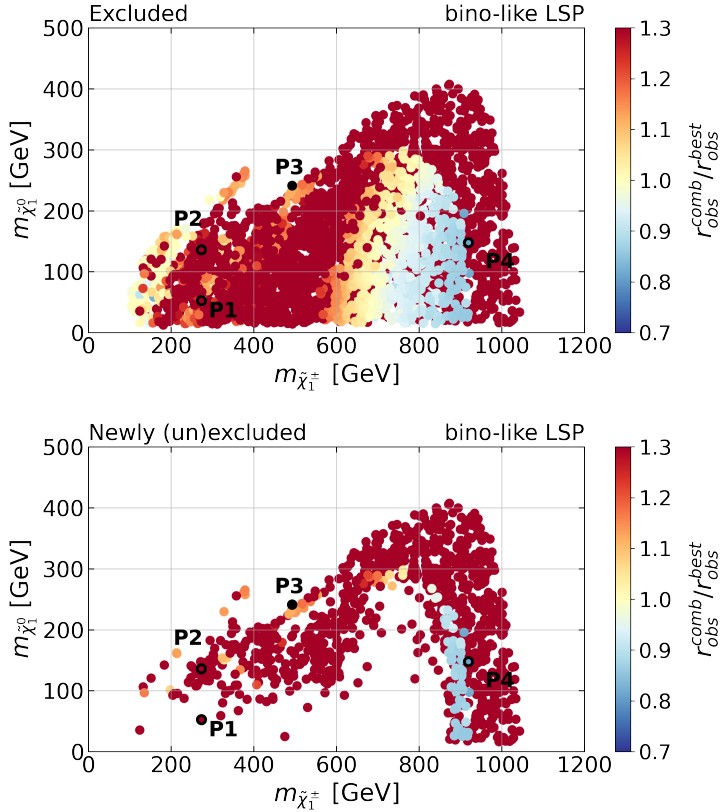

Figure 17: Impact of the combination on the exclusion power with respect to the most sensitive analysis. All the points featuring a bino-like LSP that are excluded by the combination or by the most sensitive analysis are shown on the top panel. The subset of points whose exclusion status changed with the combination is shown on the bottom panel. The circled points denote points P1, P2 and P4, for which the likelihoods are plotted in Figures 15 or 16. Point P3 (black dot) is also indicated for completeness, although it is not excluded.

signal is suppressed and even with mass compression the combined result can no longer exclude points, resulting in the second strip of blue points seen around $m_{\tilde\chi_2^\pm} \simeq 1$ TeV. In the end, (148) 1226 points are newly (un)excluded by the combination, regardless of the LSP nature.

The impact of analysis combination on the EW-ino parameters is shown in Figure 19. Dark purple points show points excluded by the combination, while light pink points show the exclusion considering only the most sensitive analysis. The top left and top right plots correspond to points mostly in the wino-bino and higgsino-bino scenarios, respectively. The bottom left plot mostly depicts points with a higgsino-like LSP, also showing up in Figure 18. Finally, the bottom right plot contains the three other plots as well as the few points with $M_2 \lesssim M_1, \mu$. We can clearly see that the gain in exclusion power at high masses only takes place in the wino-bino case, as discussed above. In the other scenarios, the gain is situated at intermediate masses. Projected onto the $M_2/M_1$ vs. $M_2/\mu$ plane, the footprint of excluded points does not increase. We recall however that the number of excluded points increases by 35% with the combination.

It is also interesting to consider which points or scenarios still escape exclusion. Points can evade exclusion due to three major reasons: i) small cross sections for large BSM masses, ii) small cross sections for higgsino production, and iii) a large number of complex topologies which are not constrained by the simplified model results contained in the database. Figure 20

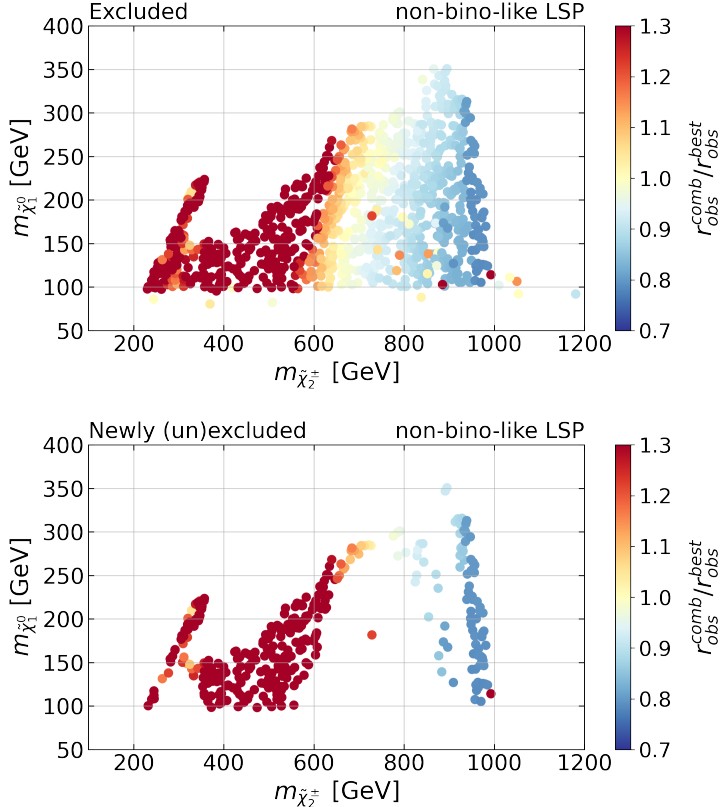

Figure 18: As Figure 17 but in the $m_{\tilde{\chi}_1^0}$ vs. $m_{\tilde{\chi}_2^\pm}$ plane, for points featuring a non-bino-like LSP.

shows the points with a bino-like LSP allowed by the combination ($r_{\text{obs}}^{\text{comb}} < 1$). The colour represents the $\tilde{\chi}_1^+$ wino content, quantified through $V_{11}$, the upper left component of the $V$ matrix defined in eq. (6). Purple points are therefore points with a wino-like NLSP, while green points feature a higgsino-like NLSP. As expected, the allowed parameter space for points with a wino-like NLSP is smaller, due to their larger production cross sections. We also see that several points avoid exclusion at low masses if $m_{\tilde{\chi}_1^0} \lesssim m_{\tilde{\chi}_1^\pm}$. These points display mixed scenarios, where the number of complex topologies is large, thus diluting the signal going into the simple 1-step decay topologies constrained by the database. The most important observation from Figure 20 is, however, that there is a sizeable region which is definitely excluded. This region extends up to $m_{\tilde{\chi}_1^\pm} \approx 900$ GeV for higgsino-like NLSP and up to $m_{\tilde{\chi}_1^\pm} \approx 1$ TeV for wino-like NLSP. Such a region does not exist for non-bino-like LSP points.

Last but not least, it is interesting to see which fluctuations exist in the data with respect to the expected background. To this end, Figure 21 shows the distribution of $r_{\text{obs}}/r_{\text{exp}}$ across the EW-ino dataset. Values below one indicate excesses in the data, while values above one signal under-fluctuations. With enough data, if there is no BSM signal, one expects $r_{\text{obs}}/r_{\text{exp}}$ to be normal distributed around unity. In Figure 21, the results for the most sensitive individual analysis show a large spread in $r_{\text{obs}}/r_{\text{exp}}$, with only a subdominant number of points having $r_{\text{obs}} \approx r_{\text{exp}}$. Overall, the over-fluctuations outweigh the under-fluctuations with 54% of points having $r_{\text{obs}}/r_{\text{exp}} < 1$. In the combination, on the other hand, large fluctuations are almost always suppressed, and the distribution of $r_{\text{obs}}/r_{\text{exp}}$ more centered towards one. Nonetheless, the tendency for excesses still remains: $r_{\text{obs}} < r_{\text{exp}}$ occurs for 72% of the points. This is even more true when considering only the points not excluded by the combination: here, $r_{\text{obs}} < r_{\text{exp}}$ occurs for 75% of the points. One should note, however, that in this case the combination

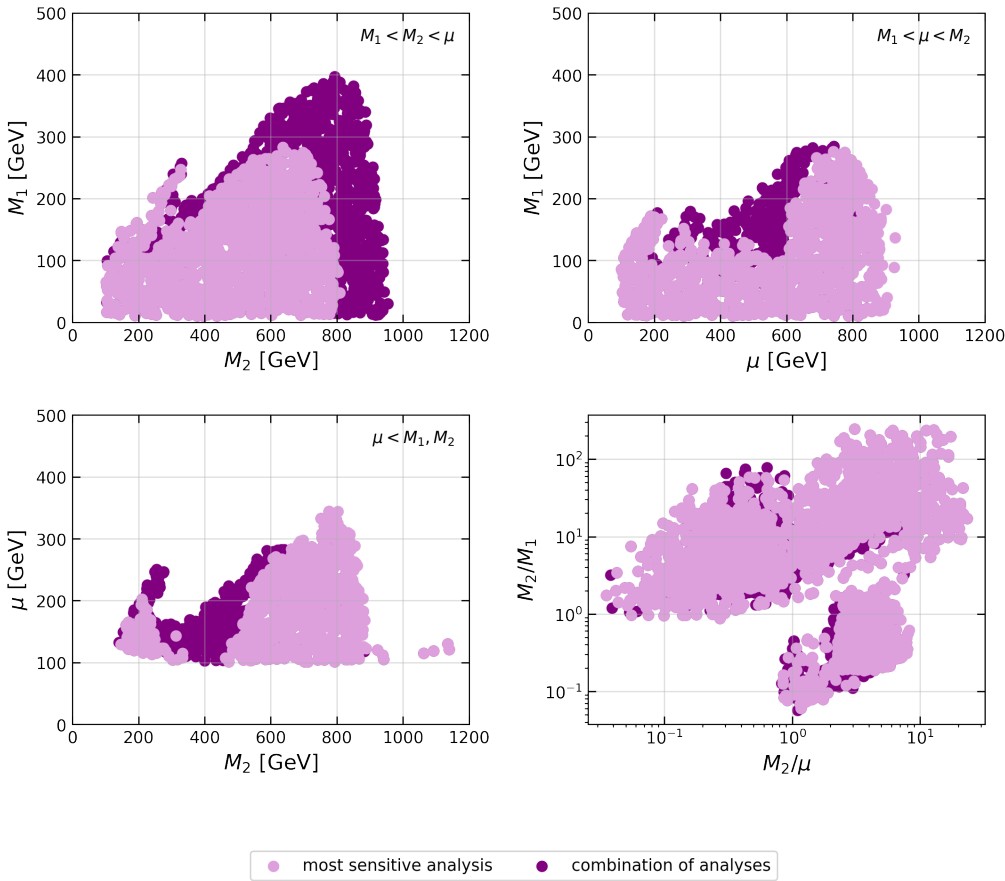

Figure 19: Observed exclusion from the most sensitive analysis and from the combination, in terms of $M_1$, $M_2$, and $\mu$. Besides mixed scenarios, the top left plot corresponds to the wino-bino scenario, the top right plot to the higgsino-bino scenario, and the bottom left plot to the wino-higgsino scenario. The bottom right plot covers all cases.

often includes only a small number of analyses, which is not necessarily sufficient to mitigate fluctuations from individual searches. It will be exciting to see how this tendency will evolve with Run 3 of the LHC.

## 6 Conclusions and outlook

The EW-ino sector of the MSSM is difficult to constrain at the LHC in a generic manner, because mixing effects lead to large variations in production cross sections and decay branching ratios. As a consequence, limits on charginos and neutralinos established in the context of simplified models do not hold in general. A reinterpretation in realistic theoretical scenarios, combining the wealth of experimental results from searches in different final states, is necessary.

In this paper, we reviewed the EW-ino sector of the MSSM and the variety of signatures that can occur depending on the scenario, even when all other SUSY particles are decoupled. We also gave an overview of the EW-ino searches from ATLAS and CMS in different channels, for which EM-type results are available in SMODELS v2.3, discussing which information is available to help the statistical evaluation (such as the procedure to combine signal regions), and which analyses do or do not overlap, based on their SRs. The latter is important for

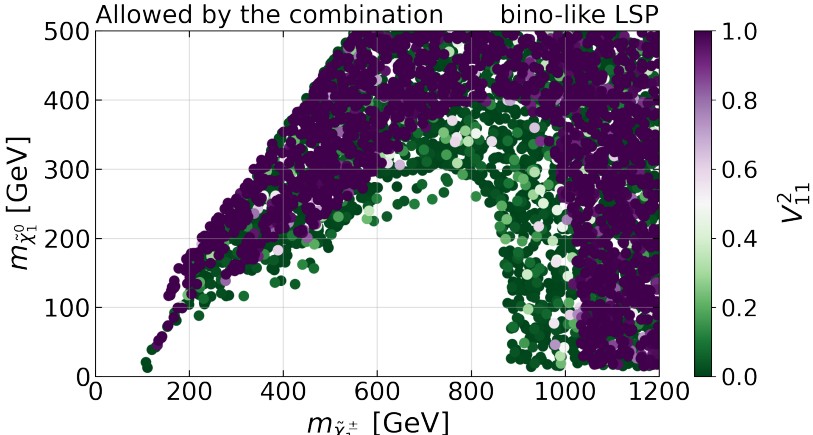

Figure 20: Points with a bino-like LSP not excluded by the combination, identified by the wino mixing of the lightest chargino. Purple points correspond to scenarios where the lightest chargino is mainly wino-like, and green points to the scenarios where it is mainly higgsino-like.

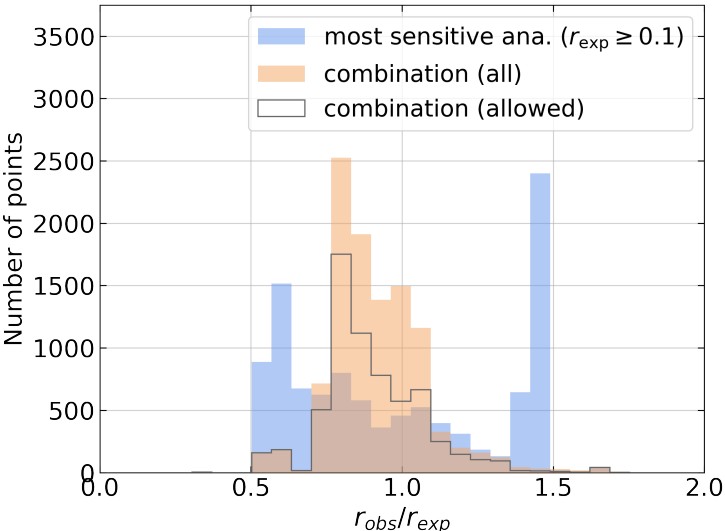

Figure 21: Ratio of $r_{\text{obs}}/r_{\text{exp}}$ from the most sensitive analysis (blue histogram), and from the combination of analyses (orange histogram) considering all scan points. Also shown is $r_{\text{obs}}/r_{\text{exp}}$ for the points allowed by the combination (grey steps). Only the points with $r_{\text{exp}} \geq 0.1$ fill the "most sensitive ana." histogram in order to have a fair comparison with the combination. If $r_{\text{obs}}/r_{\text{exp}} > 1$, it means the model point is influenced by an under-fluctuation in the data, otherwise, by an excess.

determining which analyses can be regarded as approximately uncorrelated, such that they can be combined in a global analysis. From the 16 searches considered, 13 are from Run 2, and 9 (7 ATLAS and 2 CMS) for full Run 2 luminosity. Besides the 'conventional' EW-ino searches in final states with leptons, this also comprises the new ATLAS and CMS searches for EW-inos in fully hadronic final states.

This set of analyses was then used to constrain the EW-ino sector of the MSSM in a global likelihood analysis. To this end, we confronted 18K points with promptly decaying EW-inos from a large scan over $M_1$, $M_2$, $\mu$, and $\tan\beta$ against the experimental data (points with long-lived charginos, as typical for wino-LSP scenarios, were not considered, as they are constrained by disappearing-track searches). For each point in the scan, the combination of analyses that maximises the sensitivity was determined, and the ratio of predicted over excluded cross sections (the so-called $r$-value in SMODELS) computed. The dynamic determination of the best combination is necessary, because not all searches considered in this study are approximately uncorrelated, and therefore different sets of combinations are possible. Consequently, since the sensitivity of each individual analysis changes from point to point in the scan, so does the sensitivity of any possible combination.

This approach allowed us to highlight the various most sensitive combinations and how they populate the parameter space, the effect of individual analyses on the global likelihood, and the combination's impact on the exclusion power compared to an analysis-by-analysis approach. We also showed how the combination in the high mass region is dominated by the two ATLAS and CMS hadronic searches, which recorded opposite fluctuations, leading to a fair augmentation of excluded points, while unexcluding only a few. For lighter masses, the combination's behaviour is less intuitive, especially because of the increased sensitivity to other analyses, resulting in an augmented number of analyses entering the combination. All in all, the combination of analyses increases the number of points (expected to be) excluded by (48%) 35% compared to the most sensitive analysis. More importantly, it mitigates the sensitivity to fluctuations in the data, therefore leading to more robust constraints.

One aspect that we could not cover in our study is the small excess seen in searches for compressed EW-inos by both ATLAS [42, 66] and CMS [74, 75], which can be interpreted, e.g., as the production of higgsino-like EW-inos with chargino-LSP mass splittings of roughly 5–15 GeV. Such light higgsinos are in particular motivated by naturalness arguments, see e.g. [76] and references therein. The reason that we cannot comprehensively treat this case is that only one of the relevant analyses, ATLAS-SUSY-2019-09 [42], can presently be reused with SMODELS. The other ATLAS search, ATLAS-SUSY-2018-16 [66], does provide extensive material on HEP-DATA, but the EM results of this analysis depend on the assumed scenario, and so far could not be validated in SMODELS. The two CMS searches, CMS-SUS-18-004 [74] and CMS-SUS-19-012 [75], on the other hand, do not provide any auxiliary material for reinterpretation. Efforts to recast [66, 74, 75] and produce "home-grown" EMs for SMODELS are ongoing.

Let us finally mention that the recent study [77], based on MADANALYSIS 5, analysed the consistency of the small excesses observed by ATLAS and CMS in soft-lepton EW-ino searches as well as in monojet searches in the context of light compressed higgsinos. While the best-fit points from the monojet searches were shown to be compatible with the excesses of the soft-lepton searches, also in [77] no global likelihood could be built due to missing information from the experiments. The authors however point out that a global likelihood analysis for light compressed higgsinos could be performed rapidly with SMODELS once accurate efficiency maps have been obtained for the relevant analyses. This is left for future work, as the treatment of monojet analyses in SMODELS will require SMODELS v3, which is currently in preparation.

**Data management:** The complete dataset (input SLHA and SMODELS output files) for the EW-ino scan used in this paper is available on Zenodo [78], ensuring full reproducibility of the results presented in this paper.

## Acknowledgments

**Funding information**    MMA is supported by the French Agence Nationale de la Recherche (ANR) under grant ANR-21-CE31-0023 (PRCI SLDNP). SN is supported by the Austrian Science Fund (FWF) under grant number I 5767-N. TP is supported by the Initiatives de Recherche à Grenoble Alpes (IRGA) ANR-15-IDEX-02 project no. G7H-IRG21B26 (APM@LHC). AL is supported by FAPESP grant no. 2018/25225-9 and 2021/01089-1. The work presented here was, moreover, supported in part by the IN2P3 master project "Théorie – BSMGA".

## A    Interface to resummino

In order to add electroweak (EW-ino and/or slepton) production cross sections beyond leading order to an SLHA file, smodelsTools now provides a new cross section computer based on RESUMMINO 3.1.2 [20, 21]. This allows for the computation of EW cross sections at LO, NLO, or NLO+NLL order. The usage is:

```
smodelsTools.py xsecresummino [-h] -f FILENAME
                [-s SQRTS [SQRTS ...]] [-part PARTICLES [PARTICLES...]]
                [-v VERBOSITY] [-c NCPUS] [-C CONF]
                [-x] [-p] [-P] [-n] [-N] [-k] [--noautocompile]
```

*Arguments*:

**-h, --help**      show help message and exit.

**-f FILENAME, --filename FILENAME**   SLHA file to compute cross sections for. If a directory is given, cross sections for all files in the directory are computed.

**-s SQRTS, --sqrts SQRTS**   LHC center-of-mass energy in TeV for computing the cross sections. Can be more than one value, e.g., -s 8 13 for both 8 TeV and 13 TeV cross sections; default is 13.

**-part PARTICLES, --particles PARTICLES**   list of daughter particles (given as PDG codes) to compute cross sections for. All valid combinations from the list will be considered. If no list of particles is given, the channels info from the resummino.py configuration file is used instead.

**-v VERBOSITY, --verbosity VERBOSITY**    verbosity ('debug', 'info', 'warning', 'error'); default is 'info'.

**-c NCPUS, --ncpus NCPUS**    number of CPU cores to be used simultaneously. $-1$ means 'all'. Used only when cross sections are computed for multiple SLHA files; default is 1.

**-C CONF, --conf CONF**   path to resummino.py configuration file; default is smodels/etc/resummino.py.

**-x XSEC, --xseclimit XSEC**   cross section limit in pb. If the LO cross section is below this value, no higher orders will be calculated; the default is 0.00001, set in the `smodels/etc/resummino.py` file.

**-p, --tofile**   write cross sections to file (only highest order).

**-P, --alltofile**   write all cross sections to file, including lower orders.

**-n, --NLO**   compute at the NLO level (default is LO).

**-N, --NLL**   compute at the NLO+NLL level (takes precedence over NLO, default is LO).

**-k, --keep**   do not unlink temporary directory.

**--noautocompile**   turn off automatic compilation.

To give a concrete usage example,

```
smodelsTools.py xsecresummino -s 13 -p -n -part 1000023 1000024 \
                -f test/testFiles/resummino/resummino_example.slha
```

will compute the $\tilde{\chi}_2^0 \tilde{\chi}_1^+$, $\tilde{\chi}_2^0 \tilde{\chi}_1^-$ and $\tilde{\chi}_1^+ \tilde{\chi}_1^-$ cross sections for $\sqrt{s} = 13$ TeV at NLO for the spectrum given in `resummino_example.slha` and append them to that SLHA file.[10] Note that, if neither -p nor -P is given, the output will be written to `stdout`. Additional settings, like the PDF sets to use, are taken from the RESUMMINO configuration file, `smodels/etc/resummino.py`. There, also the default threshold ($10^{-5}$ pb) for the minimum cross section is set.

Instead of providing a list of particles via the -part argument, one can also directly specify in the python configuration file which production channels shall be considered. This is done in the "channels" python dictionary, giving the respective pdg codes:

```
{
    # channels: values (as python lists) are the pdg codes of the
    # particles X,Y to be produced in the 2->2 processes p,p -> X,Y
    # names of keys are used for debugging only
    "channels" : {"1" : [1000023,1000024], "2" : [1000023, -1000024],
                  "3" : [1000024,-1000024]
    },
    # -----------------------------------
    # The limit for the NLO calculation is determined by the 'xsec_limit'
    # variable. If the LO cross secion is below this value (in pb), no NLO
    # (or NLO+NLL) cross sections are calculated.
    "xsec_limit" : 0.00001,
    # pdfs (case-sensitive): our default is cteq66, which gives results
    # close to the PDFLHC2021_40 set. Any pdf can be used, but be sure to
    # check the name on the lhapdf website before putting any value here.
    "pdfs" : {
        "pdf_lo" : "cteq66",
        "pdfset_lo" : 0,
        "pdf_nlo" : "cteq66",
        "pdfset_nlo" : 0
    }
}
```

---

[10]Pair-production of neutralinos of the same index, here $pp \to \tilde{\chi}_2^0 \tilde{\chi}_2^0$, is negligible.

The python configuration file can readily be adapted to the user's needs to include other channels (note here that names of keys are necessary, but used for debugging only). The above usage example then simply becomes

```
smodelsTools.py xsecresummino -s 13 -p -n \
                -f test/testFiles/resummino/resummino_example.slha
```

If a different RESUMMINO configuration file than the default one should be used, this can be specified with the `-C` argument. Note that options set directly on the command line always take precedence over the settings in the configuration file.

RESUMMINO needs Boost header files, the GNU Scientific Library (GSL), and the LHAPDF library. RESUMMINO and LHAPDF need not be installed separately, as the SMODELS build system takes care of that. Boost and GSL, however, should be installed by the user before compiling RESUMMINO. The requirements and installation methods are explained in detail in the Installation and Deployment section of the online manual.

## B   Impact of mass compression

In this appendix, we explain our choice for the `minmassgap` parameter, which controls mass compression in SMODELS, and its influence on the results of this study.

The decay of an intermediate state to a nearly degenerate one typically results in the generation of soft final states that are beyond experimental detection capabilities. Thus, the soft states can usually be ignored and the topology can be simplified (compressed). Given that the simplified-model results of the experimental analyses that we consider focus exclusively on direct production with a single decay in each branch, the adoption of more simplified topologies translates to an increased sensitivity, thereby enhancing the exclusion power. A detailed description of this so-called mass compression is given in [10] and in the SMODELS online documentation.

In our analysis, the non-bino-like LSP points are sensitive to the mass compression since, in such scenarios, the masses of $\tilde{\chi}_1^0$, $\tilde{\chi}_1^\pm$ and $\tilde{\chi}_2^0$ can be close to each other, as illustrated by the wino-higgsino example in Figure 2. The distributions of those points with respect to the difference between the masses of $\tilde{\chi}_1^\pm$ and $\tilde{\chi}_1^0$ (coral histogram), and the masses of $\tilde{\chi}_2^0$ and $\tilde{\chi}_1^0$ (blue histogram) are shown in Figure 22. In order to concentrate on the region where LHC results are potentially sensitive (cf. Figure 14), a cutoff of 1200 GeV on the $\tilde{\chi}_2^\pm$ mass is imposed. The portion of the $m_{\tilde{\chi}_1^\pm} - m_{\tilde{\chi}_1^0}$ histogram under the dashed red line represents wino-like LSP points. This portion is small because in wino-LSP scenarios the $\tilde{\chi}_1^\pm$ is typically long lived and, as mentioned earlier, we excluded points with a total decay width smaller than $10^{-11}$ GeV. The region under the dark red line represents the higgsino-like LSP points. The blue histogram showing the $\tilde{\chi}_2^0$–$\tilde{\chi}_1^0$ mass difference comes almost entirely from higgsino-like LSP points (recall that higgsinos correspond to a triplet of 2 neutralinos and 1 chargino). Here, the small number of wino-like points spread across the whole range of the distribution, and can barely be seen in the figure. While the $\tilde{\chi}_1^\pm$–$\tilde{\chi}_1^0$ mass difference in Figure 22 peaks below 5 GeV, many points have $m_{\tilde{\chi}_1^\pm} - m_{\tilde{\chi}_1^0} \in [5, 10]$ GeV. Moreover, the bulk of the $\tilde{\chi}_2^0$–$\tilde{\chi}_1^0$ mass difference lies between 6 and 15 GeV. We see that, with the default choice of `minmassgap` = 5 GeV in SMODELS, many non-bino-like LSP points would not be mass-compressed, which could lead to overly conservative results.

The question to consider before adjusting the `minmassgap` parameter is from which mass difference onward the decay products of an additional step in the decay chain will be hard enough to have a significant impact on the cut acceptances, so that the EMs in the database are no longer valid. Note that the efficiencies can either increase or decrease once the soft

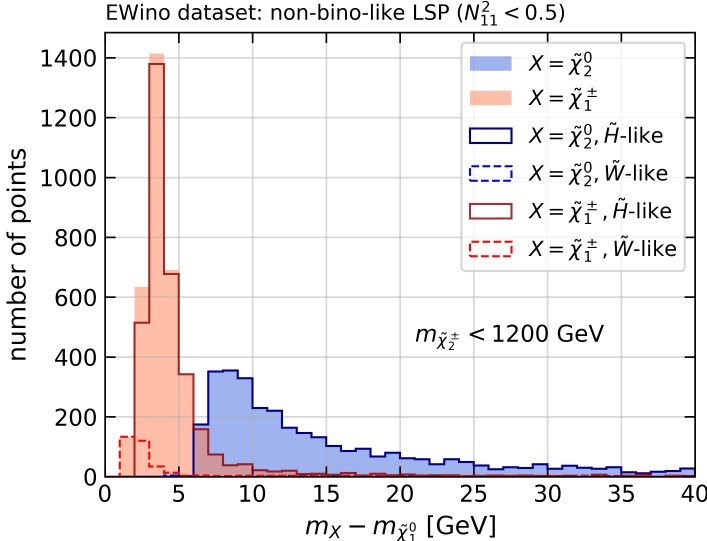

Figure 22: Number of points as a function of the mass difference between $\tilde{\chi}_1^\pm$ and $\tilde{\chi}_1^0$ (coral) and between $\tilde{\chi}_2^0$ and $\tilde{\chi}_1^0$ (blue). The plot is restricted to non-bino-like LSP points with $m_{\tilde{\chi}_2^\pm} < 1200$ GeV. The portion of higgsino-like points among the blue points is indicated by a solid dark blue line, while the points that are higgsino-like among the coral points are contoured by a solid dark red line. Additionally, the portion of the coral (blue) points that is wino-like is highlighted with a dashed red (blue) line.

particles pass a given analysis event selection. For instance, in a leptonic search that requires at least $n$ leptons and is agnostic to additional jets, the efficiencies tend to increase if the soft particles contain leptons. On the other hand, if a veto is applied on the number of leptons and/or jets, more events will be discarded and the resulting efficiency will decrease.

We studied the impact of the mass difference for a particular leptonic ATLAS search [41] (ATLAS-SUSY-2019-08), which sets limits on $W^\pm(\to \ell^\pm \nu)h(\to b\bar{b}) + \not{E}_T$ from chargino-neutralino production, by means of the MADANALYSIS 5 recast code [79]. We found that for mass differences between 5 and 10 GeV, the efficiencies are not significantly affected by the presence of additional soft states from compressed decays. However, once the mass difference was increased above 10 GeV, we noticed significant changes in the efficiencies, leading to an impact on the excluded parameter space. Thus, we choose a `minmassgap` of 10 GeV in our analysis. This ensures that more points are compressed, making them sensitive to our study, cf. Figure 22. At the same time, the associated decay products are soft enough, preserving the validity of the corresponding EMs.

The impact of changing the `minmassgap` parameter on the observed exclusions using the best combination of analyses is shown in Figure 23. The plot shows the number of excluded points with respect to the mass of the $\tilde{\chi}_1^\pm$. The SMODELS results obtained with a `minmassgap` of 5, 10, 15 and 20 GeV are shown in blue, purple, green and salmon, respectively.[11] As anticipated, the number of excluded points increases with increasing `minmassgap`, going from 3665 excluded points at 5 GeV to 4124 at 10 GeV and to 4332 (4427) at 15 (20) GeV. Moreover, the results differ only in the region of light charginos, as the mass compression only affects points for which the mass difference between the LSP and at least one other EW-ino is below `minmassgap` and $m_{\tilde{\chi}_1^0} < 500$ GeV in our dataset.

---

[11]The results for `minmassgap` = 15 and 20 GeV are shown for illustration only.

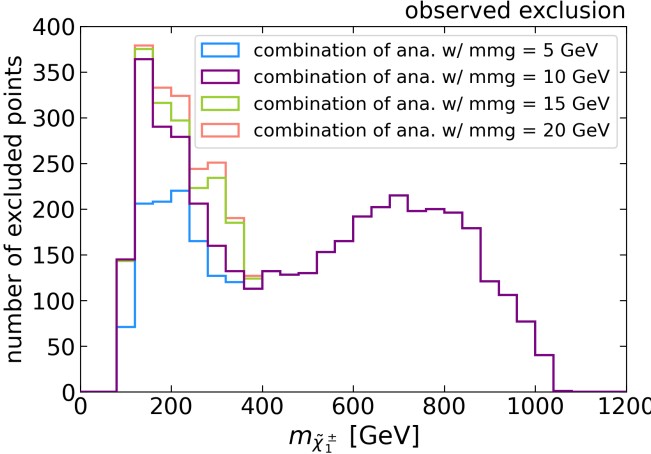

Figure 23: Observed exclusions based on the best combination of analyses. The number of excluded points is shown as a function of the mass of $\tilde{\chi}_1^{\pm}$ for different choices of the `minmassgap` parameter (abbreviated as "mmg" in the legend): 5 GeV, 10 GeV, 15 GeV, and 20 GeV.

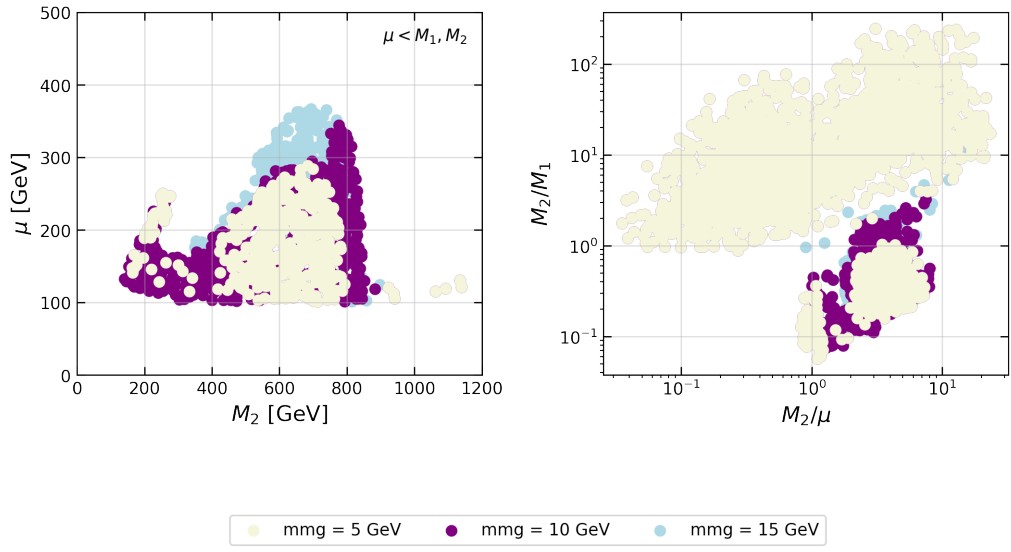

Figure 24: Observed exclusion using the best combination of analyses for various choices of the mass compression parameter (abbreviated as "mmg" in the legend), in terms of $M_1$, $M_2$, and $\mu$.

For completeness, Figure 24 shows how the observed exclusion in terms of $M_1$, $M_2$, and $\mu$ depends on the `minmassgap` parameter. Shown are the regions excluded by the combination of analyses for `minmassgap` = 5, 10, and 15 GeV, in beige, purple and light blue, respectively, in comparison with Figure 19. Since changing `minmassgap` only affects the results for a higgsino-like LSP, we only show here the case where $\mu < M_1, M_2$ in the plane of $\mu$ vs. $M_2$ (left panel) and the ratio plot of $M_2/M_1$ vs. $M_2/\mu$ (right panel). We see that, compared to `minmassgap` = 5 GeV, a value of `minmassgap` = 10 GeV gives a much better coverage of scenarios with higgsino-like LSPs, in both the low- and the high-mass regions. An extension to higher mass gaps would extend the reach to larger values of $\mu$ (see the light blue points in Figure 24) but for this, specific EMs will be needed; these are currently in preparation.

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
