# Peer review of "Global LHC constraints on electroweak-inos with SModelS v2.3"

_SciPost Physics, doi:SciPost Phys. 16, 101 (2024)_

## Round 1 · Referee Report · Anonymous (Referee 1) · 2024-2-22

Strengths

Strong paper

Weaknesses

Any substantial theory discussion beyond pmssm is avoided

Report

Report on Altakach et al., Global LHC constraints...

The paper by Altakach et al. presents a global analysis of
electroweakino pair production using 11 ATLAS and 5 CMS simplified model
seach channels. They provide a solid review of these various experimental
analyses, and then perform a scan over EWino weak scale parameter space
M1, M2, mu and tanb to present global exclusion results in the various
parameter space planes.
This paper straddles a middle ground trying to be more theoretical than
simplified models but then ignoring theory motivations such as naturalness
which is violated by too large a mu parameter. They remark at the end that
their analysis doesn't address the ATLAS and CMS soft dilepton excesses
that can arise from light higgsino pair production,
but they promise to address this case using a future version of SMODELS v3.
It may be worth noting the connection of these signals to naturalness
in the discussion of Sec. 6 since this an important motivation.
This appears to be a solid piece of work and thus I recommend publication.

---

## Round 1 · Referee Report · Anonymous (Referee 2) · 2024-2-23

Report

Report attached in PDF

Attachment

  • validity: -
  • significance: -
  • originality: -
  • clarity: -
  • formatting: -
  • grammar: -

Author:  Sabine Kraml  on 2024-03-09  [id 4353]

(in reply to Report 2 on 2024-02-23)
Category:
remark
answer to question

We thank the referee for the careful assessment and the detailed comments on our work. Most points raised by the referee have been accommodated in the revised version of the manuscript. We here supplement some additional information:

  1. Regarding the usage of the full statistical model for the ATLAS-SUSY-2018-41 analysis : Indeed a set of statistical models for the analysis ATLAS-SUSY-2018-41 has been available for a while. However, the version that we need has been uploaded on HEPData as “version 3” only in mid-November 2023, when our numerical study (which was highly CPU intensive) was basically completed. Moreover, using this full statistical model is less straightforward as it seems, as the relevant efficiency maps have to be extracted from the pyhf patchsets, and their validation turns out to be quite difficult for some of the simplified models. In particular the differences w.r.t. the official ATLAS results noted by the referee seem to remain. Likely, this is more due to the accuracy of the efficiency maps than the statistical modelling. We are investigating this in detail, but, for this study, we do not go beyond what is implemented in the v2.3 database.

  2. Regarding the impact of the NLO cross sections computed with Resummino: First of all we confirm that, for speed reasons, we computed the cross sections at NLO, not NLO+NLL, in the scan (section 4.1). Going from LO to NLO is indeed quite relevant: about 15% less points would be flagged as excluded when using only the LO cross sections. This corresponds roughly to the relative difference between LO and NLO cross sections. We have added a sentence at the end of the second paragraph of section 5 (page 21) quantifying the relevance of the NLO cross sections. However, in order not to disrupt the flow of the discussion in section 5, we refrain from adding any plots.

  3. In the minor issues, on referring to the higgsino-like states as a triplet: By triplet we mean three non-degenerate masses. This is rather common in the literature, and we like to stick to this phrasing.

For everything else, please see the "list of changes" which accompanies the revised version of the manuscript.

---

## Round 1 · Referee Report · Anonymous (Referee 3) · 2024-3-4

Report

In this work, the authors performed a global analysis of LHC constraints for the promptly decaying Electroweakinos using the public tool SModelsv2.3. The relevant Electroweakino searches at ATLAS and CMS experiments are incorporated into the SModels v2.3 database. The combined likelihood has been calculated for a particular signal process considering combinability based on approximate correlation among different experimental analyses. One of the main findings of this study is that the exclusion effect of the most sensitive analyses can be diminished if a combination of analyses is considered. The results have been exemplified with various gaugino compositions of the lightest supersymmetric particle (LSP), i.e. the neutralino. The production cross-sections are analyzed at NLO with the newly added Resummino interface.
The study is rigorous and comprehensive and the paper warrants publication.

Minor comments:
1. The sleptons are considered decoupled in this study. However, the authors may wish to comment on including searches where chargino decays via slepton which is a part of the 2lepton plus mET searches (e.g. Ref. [39]), whether this will be included in the future version or not.

2. The decay branching ratio for the lightest chargino is missing in Fig. 15 and 16.

  • validity: high
  • significance: high
  • originality: high
  • clarity: top
  • formatting: excellent
  • grammar: excellent

Author:  Sabine Kraml  on 2024-03-09  [id 4352]

(in reply to Report 3 on 2024-03-04)
Category:
remark

Thank you for the concise and very positive evaluation of our work. I wish to briefly reply to point 1. and the question whether chargino decays via sleptons will be included in a future version:

The results for slepton/sneutrino-mediated chargino decays in, e.g., Ref. [39] are given for a fixed slepton/sneutrino mass only (relative to the chargino and neutrino masses). This corresponds to a 2-dimensional slice of a 3-dimensional parameter space and cannot reliably be used for the general case, where the slepton/sneutrino mass may take any value in-between the chargino and the neutrino mass. Without at least two more 2-dimensional slices (a.k.a. mass plane), allowing us to interpolate the results to general mass patterns, or a full 3-dimensional parametrisation, these results are simply not useful for reinterpretation with SModelS and we therefore do not include them in the database. The issue has been discussed in previous publications, including the RiF report from Run2. Since it is of no direct relevance to the present study, we do not wish to re-iterate on it in this paper.

---

## Round 2 · List of Changes

• Title page: The “University of Vienna, Faculty of Physics, Boltzmanngasse 5, A-1090 Wien, Austria” (affiliation 5) has been added to the affiliation of the author Sahana Narasimha

  • Page 3, first paragraph of section 2 (referee 3): In reaction to the referee’s remark about decays via sleptons, we added the sentence: “Throughout the paper we assume that the other SUSY particles are heavier than the EW-inos, so they do not influence the phenomenology discussed here.” (Details regarding the experimental results for EW-ino decays via sleptons are discussed in the reply to the referee.)

  • Page 4 (referee 2): Following the referee’s suggestion, tan(beta)=v_2/v_1 has been changed to tan(beta)=v_u/v_d.

  • Page 5 (referee 2): We added: “To guide the eye, neutralinos are indicated in blue and charginos in red.” in the caption of Figure 2.

  • Pages 15-16 (referee 2): The fact that CMS-SUS-20-004 and CMS-SUS-21-002 both use the full Run-2 dataset has been made explicit in the first lines of the corresponding paragraphs.

  • Page 17 (referee 2): In response to the question whether we enforced a light Higgs mass of 125 GeV, we added a clarifying remark at the end of the first paragraph of section 4.1: “[The mass spectra and decay tables were computed with SOFTSUSY 4.1.11], setting m_h=125 GeV for consistency of the decay calculations”.

  • Page 21 (referee 2): We added a sentence at the end of the second paragraph of section 5 quantifying the relevance of the NLO cross sections: “Regarding the importance of using NLO cross sections, 611 points (≈ 15%) would not have been excluded by the combination had we used LO instead of NLO cross sections.”

  • Page 25 (referee 3): We added a clarifying remark regarding chargino1 decays in the caption of Figure 15: “$\tilde\chi^\pm_1$ decays are not specified as the only available decay mode is into $W + \tilde\chi_1^0$”.

  • Page 32 (referee 1): We added a sentence to the next-to-last paragraph in section 6 pointing out the connection to naturalness and providing a reference: “Such light higgsinos are in particular motivated by naturalness arguments, see e.g. [76] and references therein.”

  • Page 32 (no referee): The Cernbox link has been changed to a Zenodo link in the “Data management” paragraph.

  • Pages 36-37 (referee 2): In Figure 23, the colours of the lines for minmassgap values of 15 GeV and 20 GeV were changed to green and salmon, respectively, and the description on page 36 was modified accordingly. We hope that the colours in Figure 23 are now well distinguishable.

  • References: Added reference [76] H. Baer et. al, 2024 (see the point above regarding connection to naturalness by referee 1).

---

## Editorial Decision

published